

# Investigation of spatiotemporal variability of melt pond fraction and its relationship with sea ice extent during 2000-2017 using a new data

Yifan Ding[1,5], Xiao Cheng[1,2,5*], Jiping Liu[3], Fengming Hui[1,2,5], Zhenzhan Wang[4]

[1]College of Global Change and Earth System Science, and State Key Laboratory of Remote Sensing Science, Beijing Normal
University, Beijing 100875, China
[2]School of Geospatial Engineering and Science, Sun Yat-sen University, Zhuhai, 519000, China
[3]Department of Atmospheric and Environmental Sciences, University at Albany, State University of New York, Albany, NY
12222, USA
[4]Key Laboratory of Microwave Remote Sensing, National Space Science Center, Chinese Academy of Sciences, Beijing,
China
[5]University Corporation for Polar Research, Beijing 100875, China

*Correspondence to*: Xiao Cheng (polecx@163.com)

**Abstract.** The accurate knowledge of variations of melt ponds is important for understanding Arctic energy budget due to its
albedo-transmittance-melt feedback. In this study, we develop and validate a new method for retrieving melt pond fraction
(MPF) from the MODIS surface reflectance. We construct an ensemble-based deep neural network and use in-situ observations
of MPF from multi-sources to train the network. The results show that our derived MPF is in good agreement with the
observations, and relatively outperforms the MPF retrieved by University of Hamburg. Built on this, we create a new MPF
data from 2000 to 2017 (the longest data in our knowledge), and analyze the spatial and temporal variability of MPF. It is
found that the MPF has significant increasing trends from late July to early September, which is largely contributed by the
MPF over the first-year sea ice. The analysis based on our MPF during 2000-2017 confirms that the integrated MPF to late
June does promise to improve the prediction skill of seasonal Arctic sea ice minimum. However, our MPF data shows
concentrated significant correlations first appear in a band, extending from the eastern Beaufort Sea, through the central Arctic,
to the northern East Siberian and Laptev Seas in early-mid June, and then shifts towards large areas of the Beaufort Sea,
Canadian Arctic, the northern Greenland Sea and the central Arctic basin.

**1 Introduction**

The onset of the melting season, typically in late spring, is driven by the increase in incoming solar radiation and weather
systems originating in the mid-latitudes. As snow and ice melts, the melt water accumulates on the surface in valleys of sea
ice topography. The area and depth of the ponds depend on sea ice topography and internal structure. Field experiments showed
that melt ponds are smaller and deeper on multiyear ice (MYI), but larger and shallower on seasonal sea ice (e.g., Yackel et
al., 2000; Sturm et al., 2002; Eicken et al., 2004; Perovich and Polashenski, 2012; Polashenski et al., 2012). Melt ponds have
lower albedo than snow and bare ice (e.g., Tschudi et al., 2001; Perovich et al., 2002; Perovich and Polashenski, 2012; Webster
et al., 2015). This increases the absorption of solar radiation and further melts the snow and ice, leading to an important positive



albedo feedback (Curry et al., 1995; Perovich and Tucker, 1997; Perovich et al., 2007a). There are several steps of the evolution of sea ice albedo with melt ponds from May to September (Anderson and Drobot, 2001; Perovich et al., 2007b; Perovich and Polashenski, 2012). The albedo drops to ~0.6 in the early stage associated with snow melt, and then further decreases to ~0.32 associated with the formation of melt ponds. As the pond drainage happens, the albedo slightly increases to ~0.54. In late July to August, as melt ponds evolve steadily, the albedo decreases by ~0.0083 d-1 (Perovich and Polashenski, 2012). In this stage, the albedo of mature ponded ice is generally below 0.3 and even lower than 0.2 for seasonal sea ice (Perovich et al., 2007b; Tschudi et al., 2008; Perovich and Polashenski, 2012). Since mid-August associated with the freeze-up, the albedo gradually increases as snow falls and temperature decreases. The evolution of melt ponds is governed by complex processes, including interactions with the snow layer, drainage rates through permeable sea ice and episodic refreezing (Eicken et al., 2004; Holland et al., 2012; Polashenski et al., 2012).

Melt ponds also lead to higher fraction of solar radiation being transmitted through sea ice during the melting season. Large-scale under ice light measurements using a ROV under summer Arctic sea ice showed that the first-year ice (FYI) extensively covered by melt ponds not only allows three times as much light to pass through than that of MYI, but also absorbs ~50% more solar radiation (Nicolaus et al., 2012a, 2013). The light transmitted through the ponded ice warms sea water under the ice, and thus enhances the ice melt. This can also increase primary productivity in the upper ocean (Arrigo et al., 2012; Palmer et al., 2014). Therefore, the knowledge of the distribution and evolution of melt ponds is of great importance for understanding the Arctic energy budget.

The Arctic has experienced amplified warming and extensive sea ice retreat in recent decades (ACIA, 2004; Serreze and Francis, 2006; IPCC, 2013). The seasonal sea ice minimum has declined by 12.4% per decade for the period of 1979-2010 (Stroeve et al., 2012). The variability of sea ice extent in recent years is much stronger than before (IPCC, 2013). Arctic sea ice extent reached its historic lowest in 2012, and the decreasing rate of sea ice extent has increased in the past two decades (Comiso et al., 2008). Due to the rapid decline of Arctic sea ice, the demand for more information of its physical and thermal properties is increasing. In recent years, extensive melt ponds coverage has been observed during the melting season, which can lead to a positive albedo-transmittance-melt feedback, partly contributing to the observed sea ice loss. Recent studies suggested the amount of melt pond fraction over Arctic sea ice can be used to improve the skill of summer Arctic sea ice prediction (Schröder et al., 2014; Liu et al., 2015). Thus, accurate knowledge of variation of melt ponds in the Arctic is extremely important for understanding and predicting sea ice changes.

Although melt ponds play an important role in the Arctic climate system (as discussed above), observations of melt ponds are far from adequate for the Arctic Ocean. Some progress on the derivation of melt pond fraction using remote sensing has been noted. Yackel et al. (2000) used the Radersat-1 SAR images to examine the evolution of melt ponds in the Canadian Archipelago and found a significant correlation between the microwave scattering coefficients and the MPF. Using the Landsat-7 data, a case study was conducted to retrieve ice surface conditions in Baffin Bay in June 2000 (Markus et al., 2002). The results showed that melt ponds have different appearance relative to dry and wet ice in the Landsat true-color image, which helps to distinguish melt ponds from sea ice. The MPF derived by the Landsat 7 has a correlation of 0.68 with the aerial



observation (Markus et al., 2003). Tschudi et al. (2008) acquired the spectral reflectance of different ice types (pond, ice, snow covered ice, and open water) over three Moderate Resolution Image Spectroradiometer (MODIS) channels near Barrow, Alaska and retrieved the MPF over the Beaufort and Chukchi Seas for summer 2004 based on a mixed-pixel algorithm, which

provides important guidance to further retrieval of the pond fraction in the entire Arctic. Based on this, Rösel et al. (2012) developed a weekly Arctic-wide MPF data spanning from 2000 to 2011 using the MODIS weekly surface reflectance data. This product showed that the averaged MPF increases quickly from mid-May to late June, reaches to its peak value in late June and early July, and then decreases gradually. The maximum pond coverage was found between 80° and 88°N. However, the retrieval algorithm used by Rösel et al. (2012) depends on the prescribed spectral reflectance of surface ice types, which

can cause uncertainties due to the changing physical properties of ice, snow and melt ponds during melting season. Zege et al. (2015) developed another MPF data using the Medium Resolution Imaging Spectrometer (MERIS) data, including the radiance coefficients as well as solar and observation angles. The improvement of the algorithm is that it does not depend on the priori reflectance of different ice types (as discussed above). They used an analytical solution of sea ice surface reflection and a radiative transfer model. Although the retrieval partly avoids the problem of prior fixed reflectance, the retrieved MPF is not

accurate for dark pond fraction and only covers part of the Arctic. Some attempts have been made to obtain pond information in small scale using SAR imagery. Kim et al. (2013) conducted a helicopter based airborne survey in the northern of Chukchi Sea in summer 2011 and obtained the MPF using 0.3 m resolution SAR images. Scharien et al. (2014b, 2014a) pointed out the potential detection of MPF on FYI using co-polarization SAR data. Mäkynen et al. (2014) analyzed the pond evolution over the north area of the Fram Strait, Greenland, and Svalbard from June to August in 2009 using ENVISAT WSM images with

HH-polarization. Han et al. (2016) retrieved the MPF on MYI using TerraSAR-X dual-polarization in the Chukchi Sea and western Arctic. Li et al. (2017) used the hybrid-polarized co-polarization SAR imagery to retrieve MPF on FYI in the central Canadian Arctic Archipelago from May to June 2012. However, it is difficult to obtain Arctic-wide MPF over long periods based on SAR image due to the limitation of the data acquisition. The MPF retrieval was also conducted using AMSR-E passive microwave data (Tanaka et al., 2016). In many regions, the difference of the averaged MPF between the MODIS

retrieval and AMSR-E retrieval during 2002-2011 is less than 5% (Tanaka et al., 2016). More recently, Liu et al. (2017) tested an alternative way for the MPF retrieval in summer 2008 by using artificial neural network (ANN). The visible imagery from the National Snow and Ice Data Center (NSIDC) are used as prior knowledge for the network training. However, they only used a single data set as prior knowledge for the network training and one network to obtain the relationship with MPF, which might lead to large uncertainty.

95       As discussed previously, using a stand-alone sea ice model, Schröder et al. (2014) found a strong predictor-predictand relationship between late spring MPF and September sea ice extent. However, Schröder et al. (2014) does not corroborate their findings against observed MPF. Using the aforementioned MODIS-derived MPF, Liu et al. (2015) found that, in agreement with Schröder et al. the MPF is an effective predictor of September ice extent. However, in terms of the details, their results are contrary to some of Schröder et al.'s more nuanced points, in particular that late spring to mid-summer pond information

promises to improve the prediction skill of seasonal Arctic sea ice minimum. The discrepancy between the modeling and



observational studies might be due to the relatively short duration of the satellite-based MPF (2000-2011, 12 years) which raises the bar for significance, as well as the uncertainty of a single product of the MPF. Thus, a longer data set is needed to revisit the potential of the MPF as a predictor for summer Arctic sea ice extent.

In this study, we develop a new data set of MPF over Arctic sea ice based on a deep neural network using multi-source melt pond observations. The data spans from 2000 to the present. Using this new MPF data, we analyze the spatial-temporal evolution and variability of melt ponds as well as over different ice types for the period of 2000-2017. We also examine the robustness of the MPF as a predictor of the September Arctic sea ice extent using our data extending to 2017.

## 2 Data and method

### 2.1 Satellite data

In this study, surface reflectance from the MODIS Terra is used to derive the MPF over Arctic sea ice (MOD09A1). The data has a spatial resolution of 500 m and is available at 8-day interval. Since the MOD09A1 selects the data with low view angle, absence of clouds or cloud shadow, and aerosol loading (Vermote et al., 2015), the MPF derived from MOD09A1 is less influenced by the satellite data quality.

    Rösel et al. (2012) used three MODIS spectral bands (band 1, 2 and 3) for the retrieval of the MPF. Here we add one near-
infrared band to extend the bandwidth over 1000 nm and to include more sensitive phase changes, such as the change within snow pack from ice to liquid and vapor (Barber et al., 1992). Specifically, we use surface reflectance from four spectral bands, two visible bands (band 1, bandwidth of 620-670 nm and band 3, bandwidth of 459-479 nm) and two near-infrared bands (band 2, bandwidth of 841-876 nm and band 5, bandwidth f 1230-1250 nm). They are used as the input data to retrieve MPF over Arctic sea ice.

Additionally, sea ice concentration (SIC) from Nimbus-7 SMMR and DMSP SSM/I-SSMIS Passive Microwave Data, developed by a revised NASA Team algorithm (Cavalieri et al., 1996), is used as sea ice mask. The time series of Arctic sea ice extent is obtained from the NSIDC website, https://nsidc.org/data/seaice_index (Fetterer et al., 2017).

### 2.2 In-situ measurements

    A number of melt ponds survey have been conducted in the Arctic. In this study, we collect and use field and aerial
observations of MPF from the following sources.

    • MPFs were made during the Healy Oden Trans-Arctic Expedition (HOTRAX) by the Polar Science Center, University of Washington (Perovich et al., 2009). The HOTRAX MPF used in our study was measured at 77°-79°N and 84°-87°N on 13, 21, 29 August and 6 September 2005 through field ice survey, ice watch and helicopter photography flights. The resolution of HOTRAX data is about 57 m×70 m.



• Melt pond information was obtained during two Chinese Arctic Research Expeditions by the Dalian University of
Technology (DLUT) (Lu et al., 2010; Huang et al., 2016). The DLUT MPF used in our study was measured at 84°N and 86°N
on 20 August 2008 and 13 August 2010 through airborne images. The resolution of the DLUT data is about 98 m×67 m.

    • Melt pond observations were collected from the ice breaker RV Polarstern during the Germany Trans-Polar cruise ARK-
XXVI/3 (Nicolaus et al., 2012b, hereafter referred to as TransArc). The TransArc MPF used in our study was measured at 84°-
87°N on 13, 29 August and 6 September 2011 through the hourly observations from the bridge of the research vessel. The
resolution of the TransArc data ranges from 500 m to 1 km.

    • Melt pond information were obtained during the Arctic Research Expeditions by the Polar Research Institute of China
(Lei et al., 2017, hereafter referred to as PRIC-Lei). The MPF of PRIC-Lei used in our study was measured at 73°-88°N on 28
July, 5, 21 August 2011, 79-84°N on 12, 20, 28 August 2012, 73-79°N on 5, 13, 29 August 2014 and 73-80°N on 7 July, 4 and
20 August 2016 through the ship observation. The resolution of the PRIC-Lei data is about 1 km×1 km.

    • Melt pond statistics are obtained from the NSIDC during the summer of 2000 and 2001 (Fetterer et al., 2008). The
NSIDC MPF was measured over four Arctic Ocean sites (Beaufort Sea, East Siberian Sea, Canadian Arctic and Fram Strait)
from May to September through visible band imagery from high-resolution satellite. The resolution of the NSIDC data is 500
m×500 m.

• MPFs were observed from the Norwegian Polar Institute (NPI) during the field campaign on Arctic sea ice north of
Svalbard in summer (Divine et al., 2015; Divine et al., 2016). The NPI MPF used in our study was measured at 80-82°N on 3
August 2012 through helicopter photography flights. The resolution of the NPI data is about 60 m×40 m.

    MPF from the above six sources are used for the training of our deep neural network (as discussed in Sect. 2.3). We also
obtain the melt pond observations from Webster (Webster et al., 2015), which is used as a completely independent validation.
The Webster MPF used in our study was measured at 69-82°N on drifting FYI and MYI in the Beaufort and Chukchi Seas
from June to August for the period of 2000-2014 through air-born and in situ observations. The resolution of the Webster's
data ranges from 8 to 25 km². Figure 1 shows the locations of the above melt pond datasets. They provide a reasonable coverage
across the Arctic Ocean.

## 2.3 Method

Recently, ANN has been used to retrieve sea ice parameters (Rösel et al., 2012; Liu et al., 2017). It is shown that ANN has
the potential to learn the complex relationship between sea ice parameters and input data. Deep neural network, stemmed from
the ANN theory, can lead to better data representation learning, holding great promise to address the challenging of the melt
pond retrieval. In this study, we develop a three hidden layer backpropagation neural network to retrieve MPF.

    The structure of the network is shown in Fig. 2. The input of the network is the grid-based spectral reflectance of the four
MODIS spectral bands. 25, 35 and 45 neurons are used in the first, second and third hidden layer, respectively. The output of
the network is the MPF. For the training of the network, the spatial resolution of the input data is 500 m. In this way, the output
MPF has comparable resolution with the observed MPF mentioned in Sect. 2.2. The number of the training samples used in





the network is shown in Table 1. The network randomly selects 80%, 10% and 10% of the data as the training, validation and test data. The cost function used in our deep neural network is the mean squared error.

## 3 Results

### 3.1 Validation

First, we train the deep neural network described in Sect. 2.3 using 500 m spectral reflectance of the four MODIS spectral bands as the inputs. Here we run the network 100 times with random weights and biases assigned initially for the neurons in the network, which generates 100 networks. Figure 3 shows the performance of the 100 networks. The correlation coefficients between the observations and the MPF obtained from these networks ranges from 0.69 to 0.80. This suggests that the relationship identified by the network is quite sensitive to initially assigned weight and bias. To obtain a robust network with high correlation between the retrieved and observed MPF, we apply the ensemble averaging to the networks within 10-90 percentile of the 100 networks. The average of MPF derived from the 80 networks is used as the final output. As shown in Fig. 3, the retrieved MPF from the ensemble averaging is in a good agreement with the MPF measurements. The correlation coefficient is 0.76, which is higher than four fifth of the 100 networks. Second, we apply the relationship obtained from the ensemble averaging to the MODIS spectral reflectance remapped on a polar stereographic grid with a spatial resolution of 12.5 km to generate the MPF from 2000 to 2017.

Then, we compare our MPF retrieval and the retrieval from University of Hamburg (Rösel et al., 2012) with the observations from the aforementioned six sources that are resampled on the 12.5 km polar stereographic grid. Compared to the retrieval from University of Hamburg, we do not use the fixed spectral reflectance of each surface type (i.e., bare ice, snow cover ice, melt pond and open water) to build the relationship. Instead, we use the observed MPF from multi-sources to directly train the deep neural network. The advantage of our method is that it avoids large uncertainties of spatially and temporally varying reflectance associated with different surface species, which can result in large uncertainties for the retrieval of type fraction, i.e., the fixed cost function used in Rösel et al. (2012) based on only three surface features (pond, snow covered ice, and open water) might cause large uncertainties in the MPF retrieval.

Figure 4 shows the correlation and root mean square errors (RMSE) between the observed and retrieved MPF. The correlation coefficients of our retrieval are 0.61, 0.67, 0.33, 0.54, 0.48 and 0.55 for HOTRAX, DLUT, TransArc, PRIC-Lei, NSIDC, and NPI respectively (red bars in Fig. 4a). For HOTRAX and NSIDC, the correlation coefficients of our retrieval are comparable to those of University of Hamburg. Our retrieval has much better relationships than that of University of Hamburg for DLUT, TransArc and PRIC-Lei, in particular the retrieval by University of Hamburg shows negative correlation for the DLUT data. The RMSE of our MPF retrieval is 12%, 5%, 17%, 9%, 6% and 10% for HOTRAX, DLUT, TransArc, PRIC-Lei, NSIDC and NPI, respectively (Fig. 4b), which are also comparable or smaller than those of University of Hamburg.

It should be noted that the MPF observations from the six sources have been used as the training (target) data in our deep neural network. Thus, they are not totally independent validation data even after the resampling. To assess our retrieval,





independent MPF measurements are needed. The MPF measured from Webster is used for the evaluation, which is not used in the network. The independent validation between retrieved MPF and Webster observation covers 69-82°N from 2000 to 2014. Our retrieval shows a relatively higher correlation 0.62 and smaller RMSE 7.9% compared to that of University of Hamburg.

**3.2 Spatiotemporal variability of MPF**

Figure 5 shows the evolution of the averaged MPF relative to the ice-covered area in the Arctic from 9 May to 6 September for the period of 2000-2017. Here we only consider the grid cell with sea ice concentration greater than 15%, which is typically used as the threshold to define sea ice extent. The evolution of the MPF shows a relatively asymmetrical growth and decay. The MPF is ~8% in early May and increases to ~12% in late May. It then increases substantially in June and reaches ~23% by the end of June. After that the MPF grows very slowly, reaching to its peak in mid to late July (~26%). Since late July, the

MPF decreases gradually, reaching to ~14% in early September. We note that the interannual variability of the MPF during the decay period is much larger than that during the growth period (grey shading in Fig. 5). This might be associated with alternating thaw and refreeze due to changes of surface temperature, rainfall and snowfall that complicates the identification of melt ponds (Rösel and Kaleschke, 2011; Rösel et al., 2012; Istomina et al., 2015). In general, our result is consistent with Liu et al. (2015) based on the MPF from University of Hamburg. However, our retrieval shows that the decay of the MPF

occurs in late July, which is about half a month later than that of University of Hamburg (Note: the MPF from University of Hamburg only covers the period of 2000-2011.) Field measurements suggested that melt ponds develop primarily from early June to early July and then come into a stable and slow growth stage until early August (Perovich et al., 2007; Perovich and Polashenski, 2012). Recent observational and modeling studies (e.g., Stroeve et al., 2014) pointed out that in recent years, the melting season of the Arctic lasts longer than before, which might be in part due to that the MPF tends to maintain at a high

level for a longer period. To understand the MPF evolution on sea ice of different ages, we calculate the MPF using the EASE-Grid Sea Ice Age Version 4 (Tschudi et al., 2019). This data set provides weekly estimates of sea ice age ranges from 1st-year to 16th-year for the Arctic Ocean derived from remotely sensed ice motion and extent (Note: 5th-year ice and above, are generally considered together).

The evolution of the MPF averaged over the FYI and MYI is similar (Fig. 5). However, the MPF on MYI is much smaller

than that over FYI in early May. The MPF on FYI is larger than that on MYI throughout the melt season and the difference is ~5%. The maximum MPF can be ~27.5% on FYI and ~23.8% on MYI. As shown in Table 2, the proportion of FYI in all ice types increases as MPF increases, i.e. for all the ice type where the MPF is above 30%, the proportion of FYI is 78%, much larger than that of other ice types. As shown in Fig. 6, in May, the growth rate of the MPF on FYI is greater than that on MYI, which is partly due to that the albedo on MYI decreases slower than that on FYI during the early stage of the ice melting

(Perovich et al., 2002). The greatest growth rate for both FYI and MYI occurs in June, although the rate of MYI is slightly faster than that of FYI, which is consistent with the result of Popović and Abbot (2017). They found that the ponds grow slower on smoother ice (the surface of FYI is smooth and that of MYI is rough) with freeboard sinking roughly proportional to the



square of the ice roughness and enhanced melting increasing linearly with the roughness. The growth rate of MPF reaches to its minimum in July due to the increased depth.

Figure 7 shows time series of the MPF anomaly averaged from 9 May to 6 September. Clearly, the MPF has large year-to-year variability, and the interannual variability after 2007 appears to be larger than that before 2007. The year 2012 (the record lowest sea ice extent) has the largest positive MPF anomaly (4.0%), and the year 2007 (the second lowest sea ice extent) also has large positive MPF anomaly. There is an increasing trend of the MPF from 2000 to 2017, but it is not statistically significant (Fig. 7). This is generally consistent with the study of Zhang et al (2018), which found an insignificant trend of annual MPF

over 1979-2016 from model simulation. The MPF over FYI and MYI tends to covary, except the year 2008 and 2015. Both of them do not show apparent trend over the 18-year studying period.

We further calculate the trend of the MPF for each individual time interval for the studying period. As shown in Fig. 8, the MPF exhibits small-to-moderate negative trends from early June to mid-July, but they are not statistically significant. By contrast, the MPF has moderate-to-large positive trends from late July to early September, and a large portion of the trends is

significant. The negative trends could be partly explained by the increasing pond drainage owing to the thinner and more porous sea ice, which reduces the loading capacity of the melt water on the surface and therefore reduces the MPF (Zhang et al., 2018). The increased MPF since late July enhances the positive albedo-transmittance-melt feedback associated with melt ponds, which partly explains why we see more record low sea ice extent in recent years. Further analysis suggests that the significant positive trends from late July to early September are largely contributed by the FYI (Fig 8b).

Figure 9 shows the spatial distribution of the MPF trend from 2000 to 2017. For the entire melting season (early May to early September), the trends in the MPF exhibit strong regional variations, and are not significant in most regions (Fig. 9a). Only some small clusters of significant trends, though scattered, are found, i.e., positive trends in the southern Chukchi, Beaufort, Kara Seas, and negative trends in the Davis Strait, the western Greenland Sea. However, for late July to early September, the Arctic Ocean is dominated by increasing trend, and most of the trends are significant.

Next, we analyze the detailed spatial evolution of the MPF from early May to early September in 2004 (the year with the second smallest MPF anomaly) and 2012 (the year with the largest MPF anomaly) using the weekly data (Fig. 10 and Fig. 11). In the first three weeks of 2004, the MPF in the range of 15-25% occurs mainly in the marginal ice zone of the Chukchi, Kara and Barents Seas, and the Baffin Bay. Melt ponds grow slowly from early to late May in 2004. Since the fourth week (June 1), intense surface melting results in substantially increased MPF (greater than 20%) in the Beaufort, Chukchi and East Siberian

Seas. In late June, large MPF (>30%) appears in the Beaufort Sea as well as most of the ice edges and extends northward into the central Arctic Ocean. In July and August, the largest MPF (>40%) is mainly confined in a band forming an arc around the periphery of the Arctic Basin extending from north of Alaska to north of western Siberia. In 2012, the MPF during the first three weeks is generally similar to that in 2004, but higher MPF (>40%) appears in the Bering Strait. Different from the evolution in 2004, the fastest growth of melt ponds starts from the East Siberian, Laptev and Kara Seas and then from the

Beaufort Sea since the fourth week. In late June, the MPF above 20% occupies almost the entire Arctic. In July and August, intense surface melting (MPF above 40%) not only appears in the narrow band as shown in 2004, but also spreads to wider





areas from the edges of the Beaufort, Chukchi and East Siberian Seas to the central Arctic. Moreover, the MPF on MYI in 2012 is obviously larger than that in 2004. The averaged MPF of early May to early September is 16.8±5.9% in 2004, 23.2±6.8% in 2012, and 19.2±5.8% averaged for 2000-2017. The higher MPF in 2012 leads to a larger positive albedo-transmittance-melt
feedback, which plays an important role in contributing to the record low sea ice extent in September 2012.

**3.3 Relationship with Arctic sea ice extent**

Recent research has pointed out that the MPF could be a potential predictor for the seasonal minimum Arctic sea ice extent (Schröder et al., 2014; Liu et al., 2015). However, the results from the modeling and observational analyses show that the predictive skill of September sea ice extent is strongly associated with the MPF during different periods. The modeling study
(Schröder et al., 2014) suggested that the MPF in May has the strongest relationship with September sea ice extent, whereas the observational study (Liu et al., 2015) indicated that a significantly strong relationship emerges as the MPF is integrated from early May through late June. The observational study is based on the MPF product from University of Hamburg for the period of 2000-2011. The question is the discrepancy between the modeling and observational studies might be due to that 1) the duration of the previous published pond data is relatively short (12 years), raising the bar for significance, and 2) the
uncertainty of the single dataset of MPF, possibly causing the inaccuracy for prediction. Here we repeat the analysis in Liu et al. (2015) using our MPF data set for the period of 2000-2017 (18 years).

Following the approach of Liu et al. (2105), we generate the detrended MPF time series by integrating the MPF relative to the ice-covered area over time and space. First, the integrated MPF in each cell is computed from 9 May to 17 May, and up to 6 September. Then, the correlation is computed between the integrated MPF in each cell and the total sea ice extent in
September. Figure 12 shows spatial map of correlation coefficients. The ice- covered Arctic ocean only shows sparse significant correlations when MPF is integrated from early May to late May (Fig. 12a). Concentrated significant correlations (>95% confidence level) first appear in a band, extending from the eastern Beaufort Sea, through the central Arctic, to the northern East Siberian and Laptev Seas, when MPF is integrated to the early-mid June (Fig. 12b), and then shifts towards large areas of the Beaufort Sea, Canadian Arctic, the northern Greenland Sea and the central Arctic basin, when MPF is integrated
to late June (Fig. 12c). Further integration to July does not change the spatial pattern of significant correlation much, although significant correlations extend into the Chukchi and East Siberian Seas which is not shown in the result of Liu et al. (2015) (Fig. 12d).

Based on the spatial pattern of correlation on 26 June (>99% confidence level), we compute the correlation between the detrended September Arctic sea ice extent and the detrended MPF integrated from 9 May to 17 May, and up to 6 September.
As shown in Fig. 13, the correlation increases quickly from the mid May to the mid-June, reaching the highest on 18 and 26 June (r=-0.94). This is consistent with that of Liu et al. (2015), which shows high correlation when MPF is integrated to 26 June, although our correlation is relatively higher than that of Liu et al. (2015). Further integration of MPF has minimal change on the correlation, and the high correlation maintains until early September. Unlike Liu et al. (2015), our analysis does not show degraded correlation from early to mid-July. Previous research indicated that pond drainage usually occurs in June and



thus the growth of melt ponds in July is generally stable (Perovich et al., 2002, Polashenski et al., 2012). The aforementioned degraded correlation is more likely caused by incorrect pond information of the University of Hamburg data set.

Following Liu et al. (2015), a linear regression is used to reproduce September Arctic sea ice extent using our MPF data. We compute the detrended September sea ice extent from the MPF integrated from 9 to 17 May, and up to 6 September during 2000–2017 (Fig. 14a). The results show that the MPF integrated from early May to the end of May fails to reproduce year-to-

year variability of September sea ice extent, which confirms the result based on the data from University of Hamburg for the period of 2000-2011. Both of the MPF integrated to the mid-June and the late June can well reflect the observed variability of September sea ice extent, particularly the extremely low ice extent in 2012 (Fig. 14a). This is also indicated by the regression error (RMSE, Fig. 14c), which decreases significantly from mid-late May to late June, reaching to 0.18 million km$^2$ on 18 June and 26 June, which is smaller than the standard deviation (0.81 million km$^2$) of the observed September sea ice extent during

2000-2017. The low RMSE maintains till late August (Fig. 14c).

Next, we evaluate the prediction skill of September sea ice extent based on the MPF information. We use the data during 2000-2008 to train the linear regression model, and then apply it to predict September sea ice extent from 2009 to 2017. The prediction skill is estimated by $1 - {\theta_f^2}\big/{\theta_o^2}$, where $\theta_f^2$ is the variance of the forecast error and $\theta_o^2$ is the variance of the detrended observed September sea ice extent (0.53 million km$^2$ for 2009–2017). For the prediction, both of the MPF integrated to the

mid-June and to the end of June show good capability of predicting September sea ice extent for most years during 2009-2017, i.e., the extremely low ice extent in 2012 can be successfully predicted, particularly when the MPF is integrated to the end of June (Fig. 14b). The prediction error decreases quickly from early May to mid-June, reaching the smallest RMSE ~0.12 million km$^2$ on 26 June, which is much smaller than the standard deviation of the detrended observed September sea ice extent. After that the errors slightly increase to ~0.19 million km$^2$ in early September (Fig. 14d). The forecast skill improves significantly

from mid-late May to early June (0.88) and maintains good skill from mid-June to mid-July. The highest predictive skill (0.95) occurs as the MPF is integrated to 26 June (Fig. 14d). Thus, our study re-confirms the predictive skill of MPF to September sea ice extent as the data extended to 2017.

**4 Discussion and Conclusion**

In this study, we develop a new method for retrieving melt ponds fraction over sea ice on the Arctic-wide scale. We develop

an ensemble-based deep neural network and use MODIS spectral reflectance from four bands as the input data and field observations of MPF from multi-sources as the target (training) data. Assessment of the target and independent MPF measurements suggests that our ensemble-based deep neural network is robust, which is proved by high correlation between the retrieved and observed MPF. Built on this, we create a new data set of MPF from 2000 to the present, which is the longest MPF data in our knowledge (Note: the MPF from University of Hamburg only covers the period of 2000-2011.). We are

attempting to make the data in near-real time and available to research community.





We analyze the spatial and temporal variability of MPF using this new data set for the period of 2000-2017. We find that the averaged MPF from early May to early September has an upward trend for the period of 2000-2017, even though it is not statistically significant. However, the MPF does show significant positive trends from late July to early September, which is largely contributed by the MPF over FYI. As mentioned previously, the FYI extensively covered by melt ponds allows much
more sunlight to pass through the ice and absorbs much more solar radiation than that of MYI (Nicolaus et al., 2012a, 2013). Spatially, in contrast to strong regional variations of the MPF trend, the Arctic Ocean is dominated by significant increasing trend during late July to early September. Thus, the significant increased MPF during late July to early September enhances the positive albedo-transmittance-melt feedback and thus might be responsible for more record low sea ice extent in recent years.

The analysis based on our MPF data during 2000-2017 further confirms that the integrated information of the MPF does promise to improve the prediction skill of seasonal Arctic sea ice minimum. This is similar to the result of Liu et al. (2015) based on the University of Hamburg data during 2000-2011. However, our data shows that the significant correlations between the integrated MPF and September sea ice extent first appear in the East Siberian and Laptev Sea in May and June and then gradually shift to the Beaufort and Chukchi Sea.

We know that clouds can interfere for obtaining a good surface reflectance (i.e., MODIS). To make sure that the strong relationship between the MPF and September sea ice extent arises from the melt ponds, rather than cloud cover which might be wrongly identified as melt ponds, here we further examine the relationship between the area of clouds in the Arctic Ocean from early May to early September and September sea ice extent during the melting season. Following the same procedure applied to the calculation of MPF as mentioned previously, the area of clouds is defined as the sum of the product of the cloud
fraction and the area of the grid cell using the Level-3 MODIS Atmosphere Eight-Day Global Product (Hubanks et al., 2018). We calculate correlation coefficients between the de-trended time series of the integrated the area of clouds at each grid cell and the de-trended time series of September sea ice extent. As shown in Fig. 15, the correlation varies between -0.2 and 0.1, and there is no significant relationship between the cloud cover and September sea ice extent during the melting season. Previous research also suggested that the clear skies with negative cloud anomaly in 2007 does not have the substantial
contribution to the extremely low sea ice extent in the same year. (Schweiger et al., 2008).

As mentioned previously, a few efforts have been taken to generate the Arctic wide MPF from the MODIS and MERIS data. However, these data are only available until 2011 (see https://icdc.cen.uni-hamburg.de/1/daten/cryosphere/arctic-meltponds.html and https://seaice.uni-bremen.de/melt-ponds). Here we create a new data set of MPF from 2000 to present, which is the longest MPF data in our knowledge. We are making the data available to research community, which can be used
to evaluate melt ponds parameterizations using in sea ice models. So far, the MPF has not been assimilated into sea ice models yet. The assimilation of melt ponds in dynamical models may improve the prediction of the Arctic sea ice minimum.

*Data availability.* The request of melt pond fraction data please contact the first author.



*Author contribution.* YD developed and analyzed the data; YD and JL wrote the manuscript; XC and JL designed the experiments, investigated the results and revised the manuscript; FH and ZW discussed the results. All authors provided
substantial input to the interpretation of the results.

*Competing interests.* The authors declare that they have no conflict of interest.

*Acknowledgements.* This research was supported by the National Key R&D Program of China (2018YFA0605901), and the China Scholarship Council (201806040111). We thank NASA EOSDIS LP DAAC for providing the MOD09A1 data and NSIDC for providing the Ice Age data. We also thank Donald Perovich at the Cold Regions Research and Engineering
Laboratory for providing the HOTRAX data, Lu Peng at the Dalian University of Technology for providing the DLUT data, Marcel Nicolaus at the Alfred Wegener Institute for Polar and Marine Research for providing the TransArc data, Ruibo Lei from the Polar Research Institute of China for providing the PRIC-Lei data, NSIDC for providing the NSDIC data, Mats Granskog and Dmitry Divine at the Norwegian Polar Institute for providing the NPI data and Melinda Webster at the Applied Physics Laboratory, University of Washington, for providing the Webster data.

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



**Table 1.** Selected samples of observed MPF from multi-sources for network training.

| Sources | Total samples | Resolution |
|---|---|---|
| HOTRAX | 26 | 57m × 70m |
| DLUT | 206 | 98m × 67m |
| TransArc | 9 | 500m to 1km |
| PRIC-Lei | 120 | 1km × 1km |
| NSIDC | 2258 | 500m × 500m |
| NPI | 164 | 60m × 40m |






**Table 2.** Percentage of the MPF relative to ice-covered area on different ice types.*

| Ice Type | 0<MPF≤10% | 10<MPF≤30% | MPF>30% |
|---|---|---|---|
| FYI | 0.54 | 0.57 | 0.78 |
| SYI | 0.17 | 0.16 | 0.09 |
| TYI | 0.10 | 0.09 | 0.05 |
| Older ice | 0.19 | 0.18 | 0.09 |

*FYI: First-year ice (0-1 years old), SYI: Second-year ice (1-2 years old), TYI: Third-year ice (2-3 years old), Older ice: Fourth-year ice and above. If the SIC of a grid remains at or above 15% throughout the melting season, then that parcel is assumed to have survived the summer minimum sea ice extent, and the parcel's age is incremented by one year.






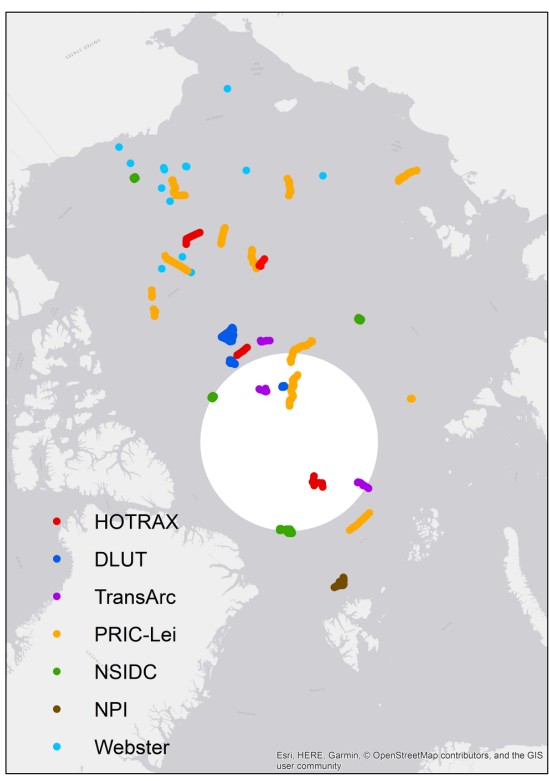

**Figure 1:** The locations of the observed MPF from multi-sources. © OpenStreetMap contributors 2019. Distributed under a Creative Commons BY-SA License.





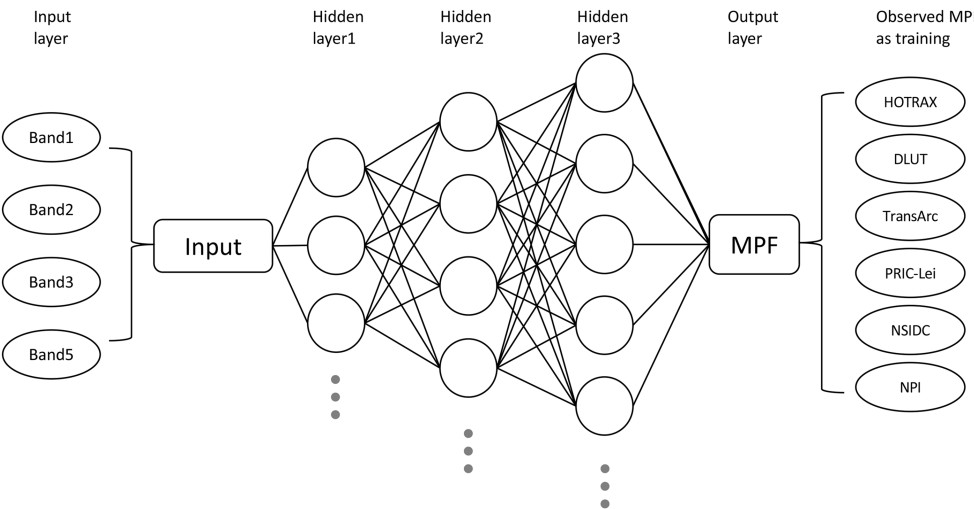


**Figure 2:** The topological configuration of the three hidden layer backpropagation neural network for the MPF retrieval.





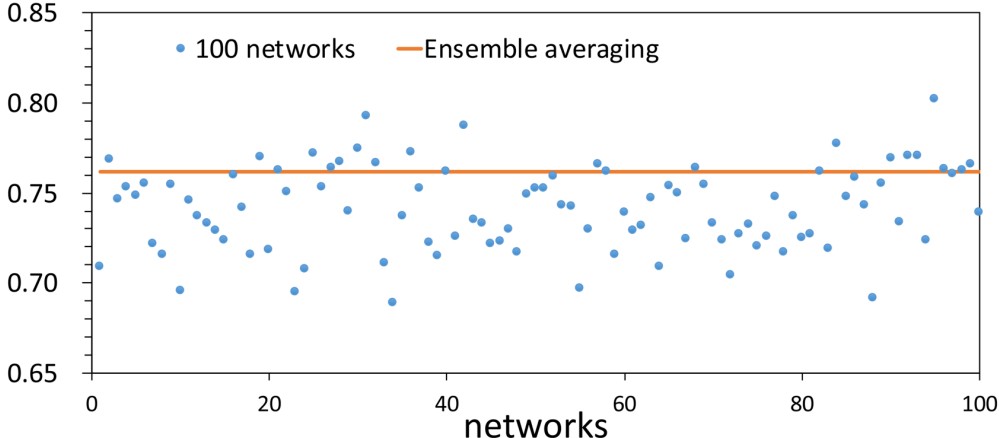

**Figure 3:** Correlation coefficients between the MPF from 100 networks and the network after the ensemble averaging and the observed MPF.



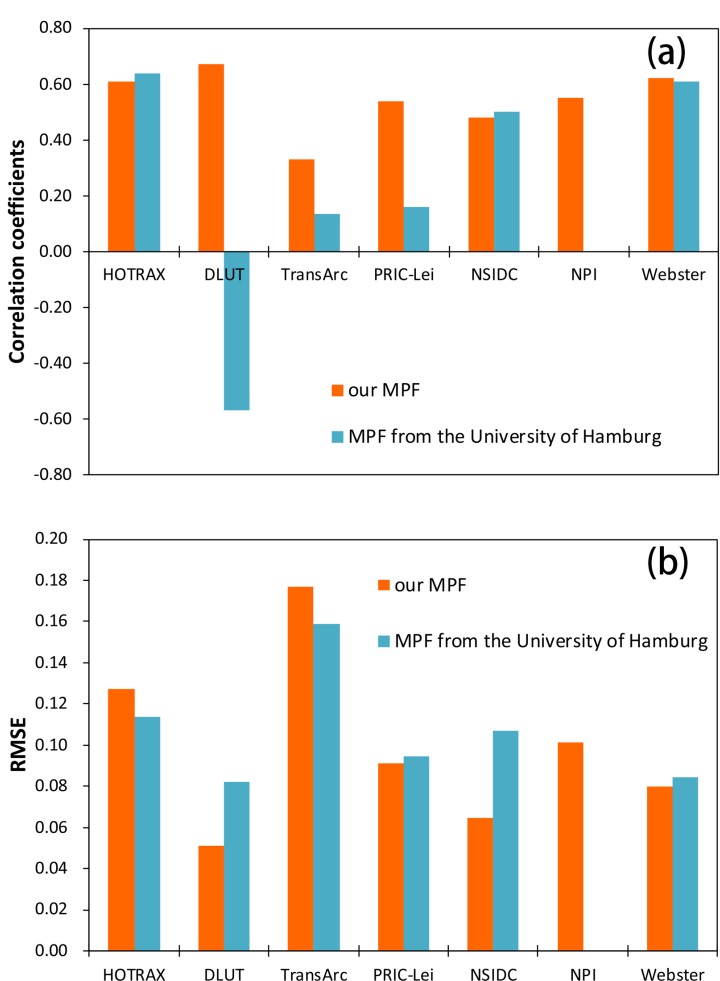

**Figure 4:** Validation of our MPF data and the MPF from the University of Hamburg against the observed MPF: (a) correlation coefficients and (b) RMSE.




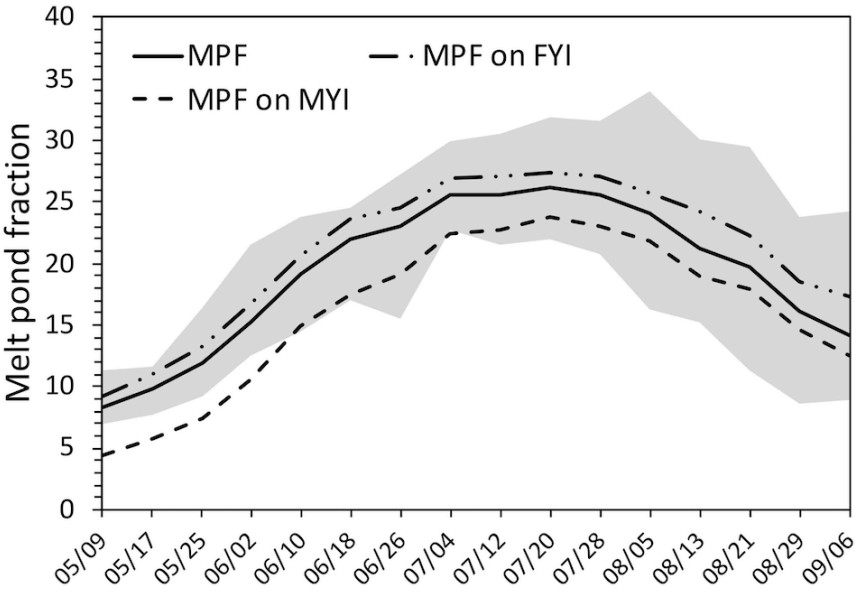

**Figure 5:** The evolution of the averaged MPF relative to the ice-covered area from 9 May to 6 September during 2000 to 2017. The grey area is year-to-year variability of MPF for the 18-year data.

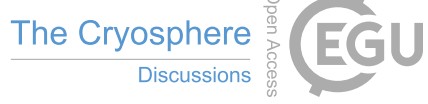

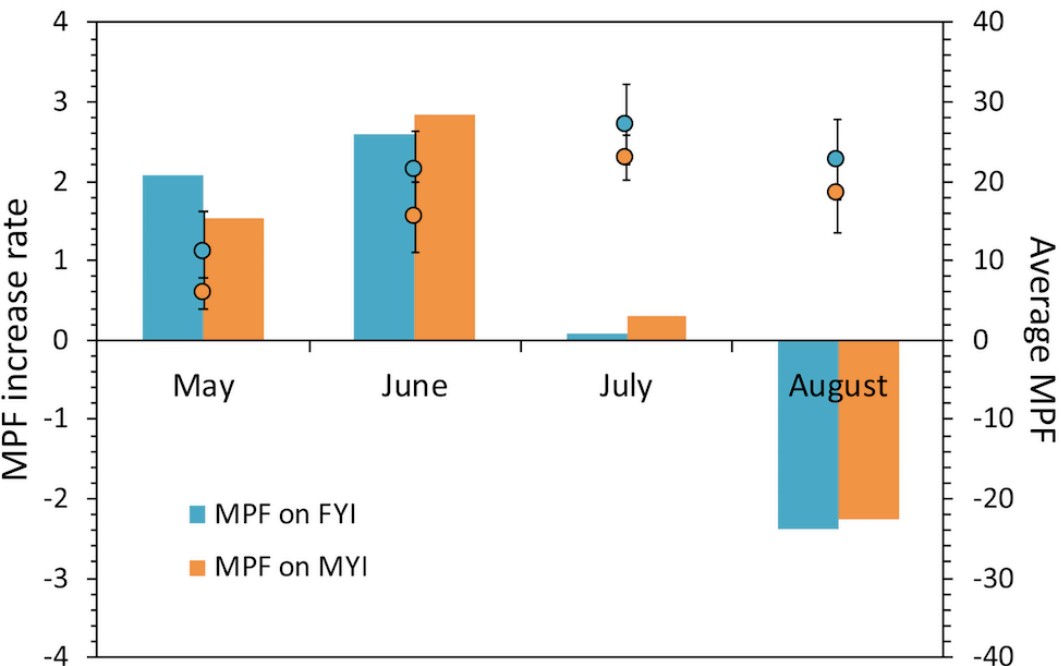


**Figure 6:** The growth rate of the MPF (bars, left y-axis) and the average MPF (dots, right y-axis) relative to the ice-covered area on FYI and MYI from May to August. The error bars represent the standard deviation.



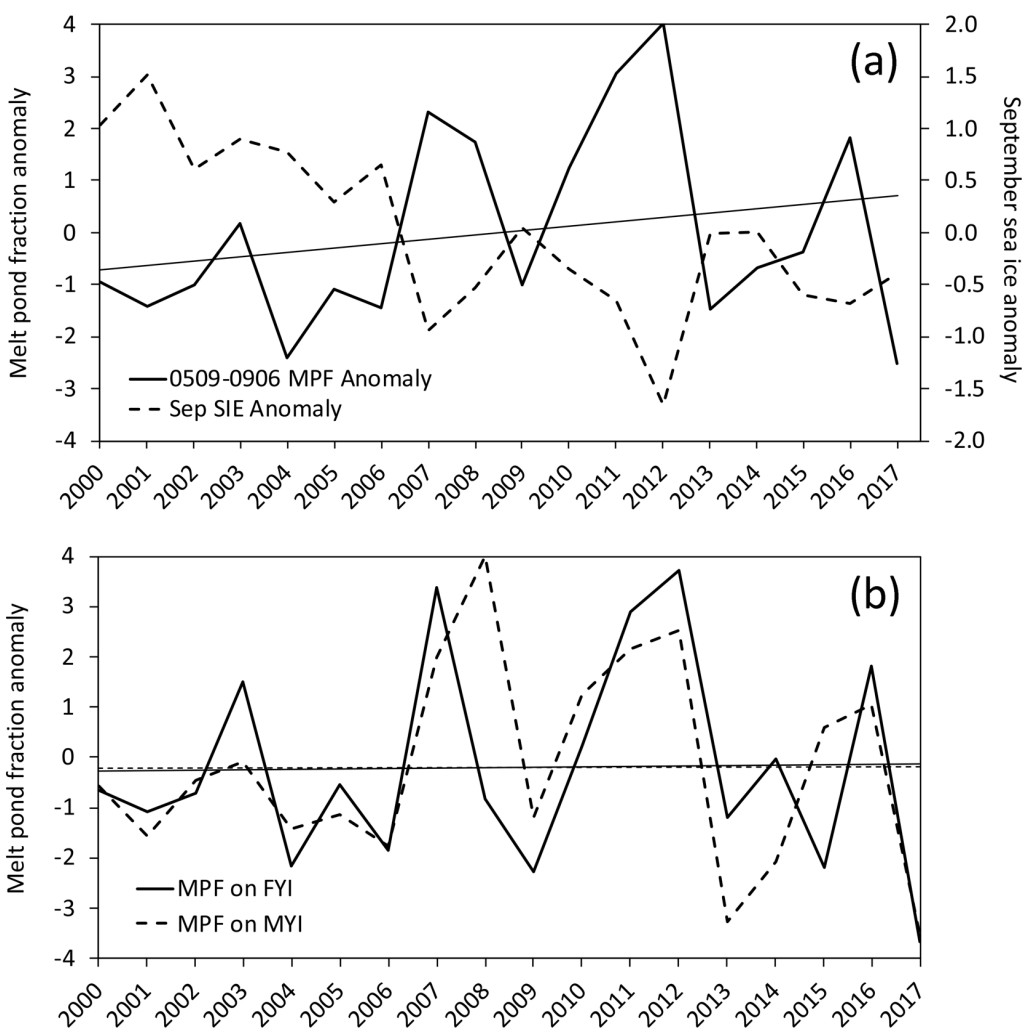

**Figure 7:** Time series of the averaged MPF anomaly relative to the ice-covered area from 9 May to 6 September. (a) the anomaly of MPF (thick solid, left y-axis) on all ice types and the anomaly of September ice extent (dashed, right-y axis). (b) the anomaly of MPF on FYI (thick solid) and MYI (thick dashed). The thin solid line in (a) is the trend of the MPF. The thin solid line and dashed line in (b) are the trends of the MPF on FYI and MYI, respectively.



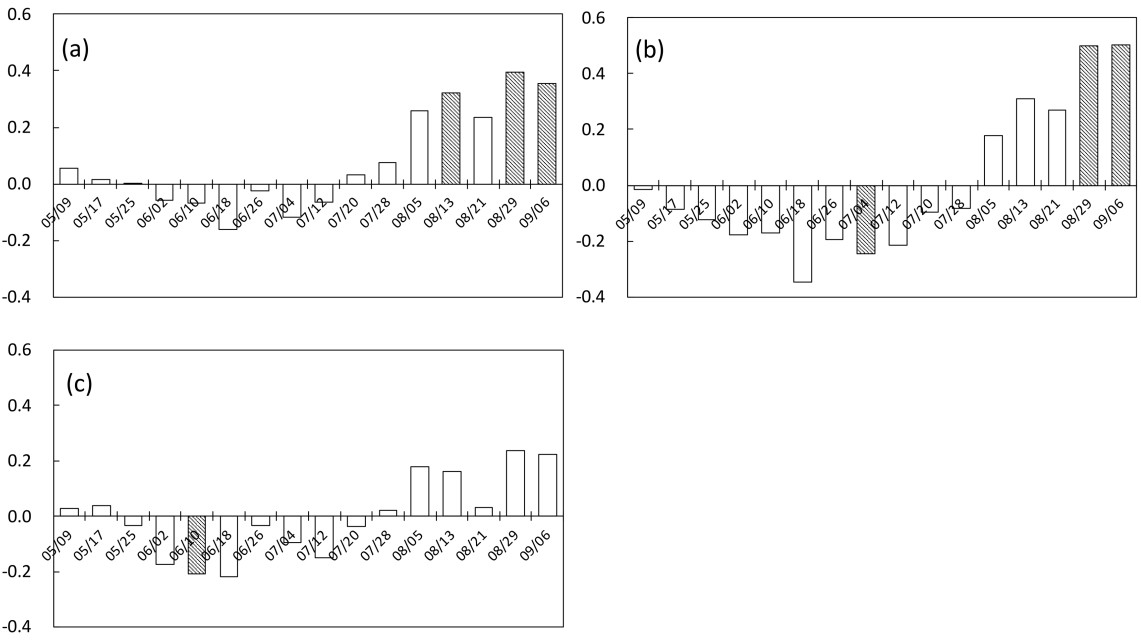


**Figure 8:** The trends of MPF relative to the ice-covered area for each individual time interval during 2000 to 2017: (a) all ice types, (b) FYI and (c) MYI. The shaded bars are above the 90% confidence levels.



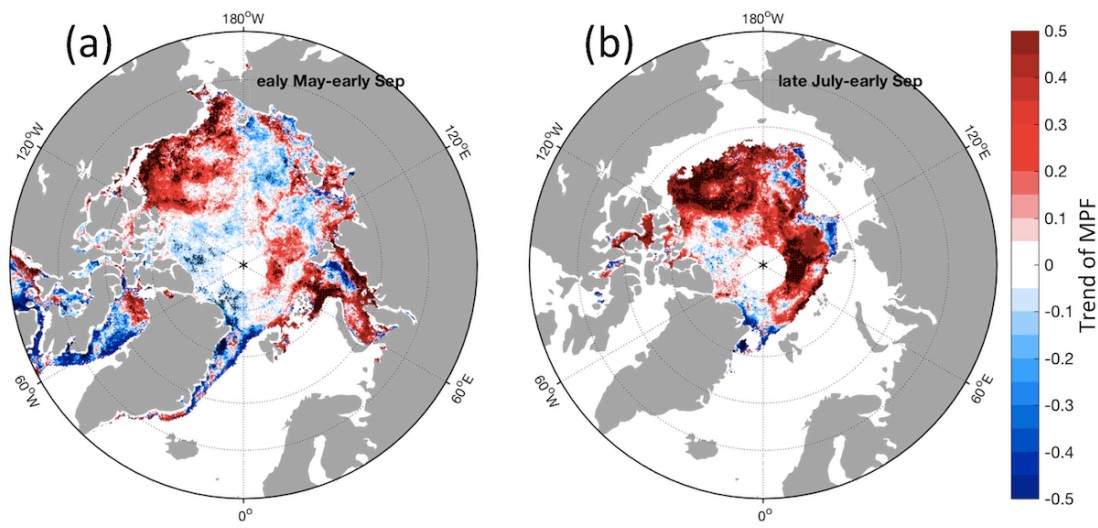


**Figure 9:** The spatial distribution of trends of the MPF relative to the ice-covered area for the period of 2000-2017. (a) the averaged MPF from 9 May to 6 September and (b) the averaged MPF from 20 July to 6 September. The dark dots represent the statistically significant trends (> 90% confidence level).



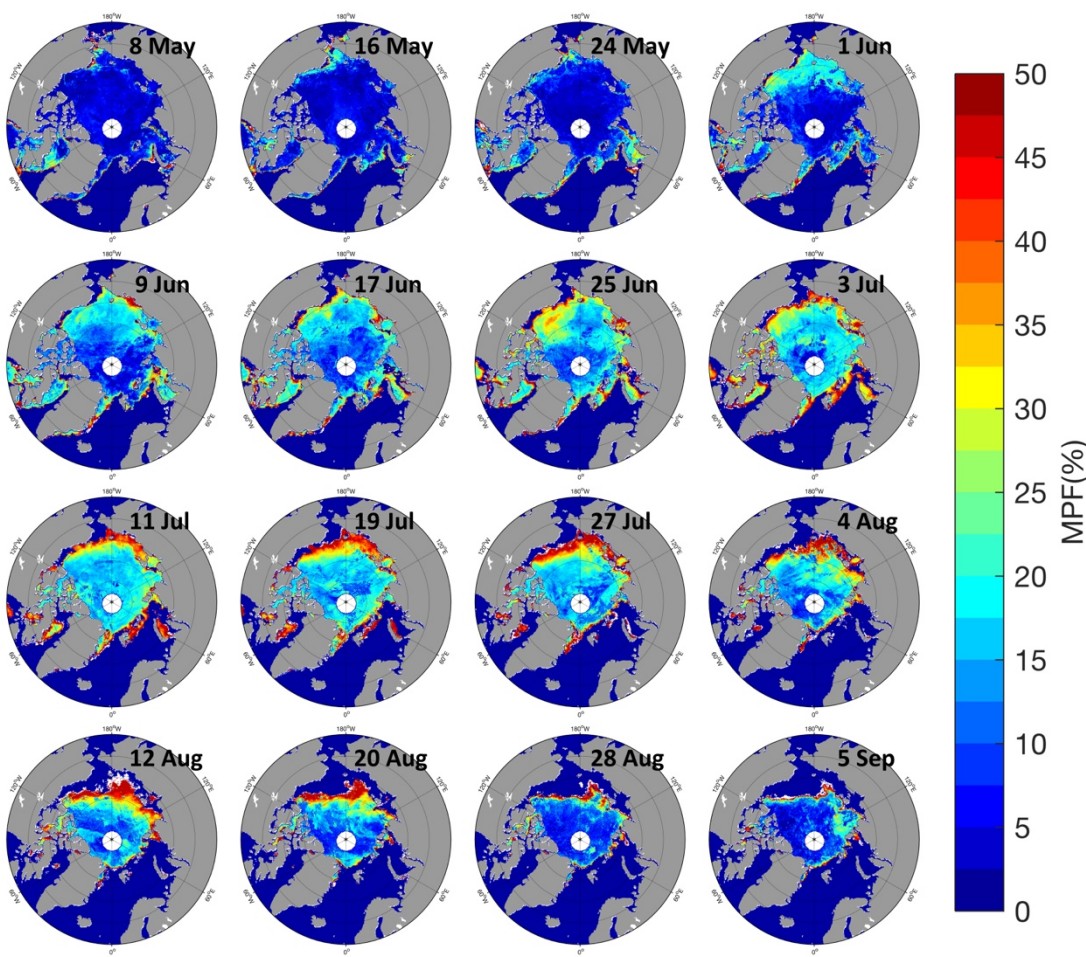


**Figure 10:** The evolution of the MPF relative to the ice-covered area from early May to early September in 2004.



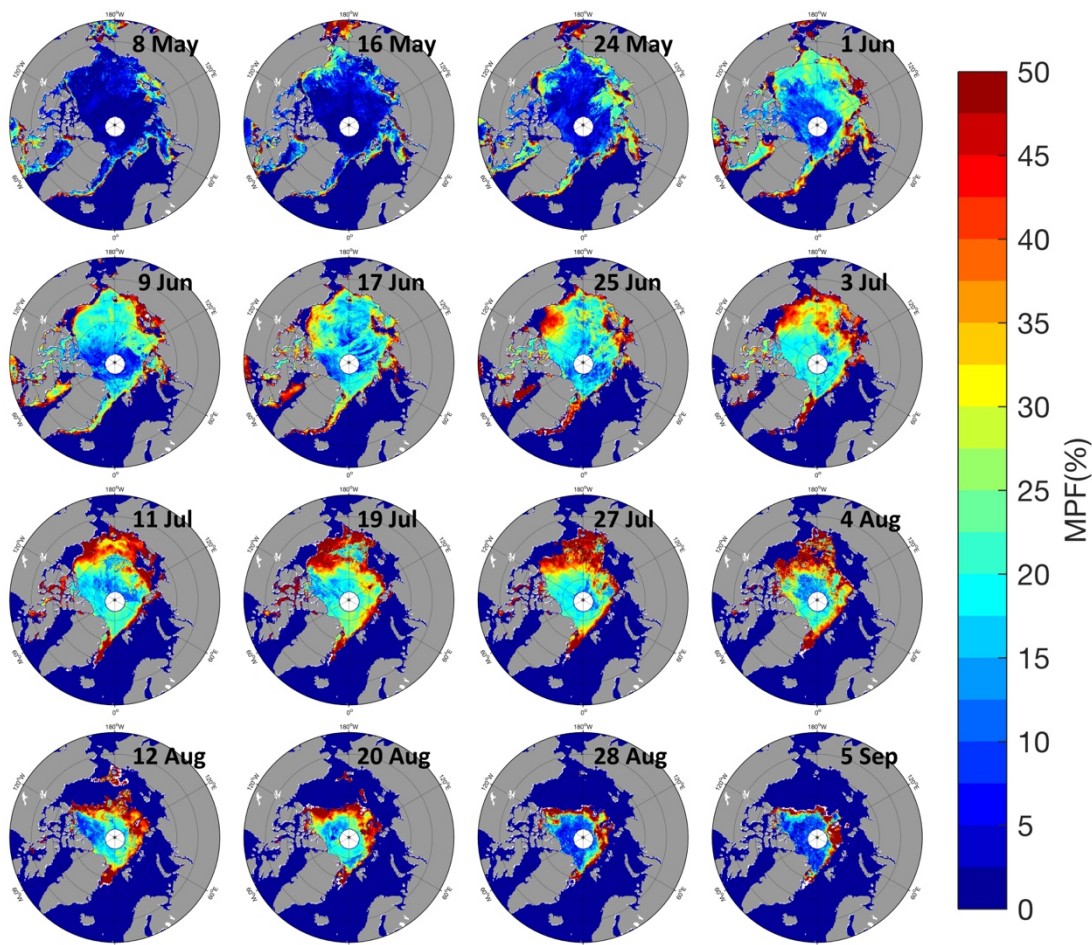

**Figure 11:** Same as Figure 10, except for 2012.




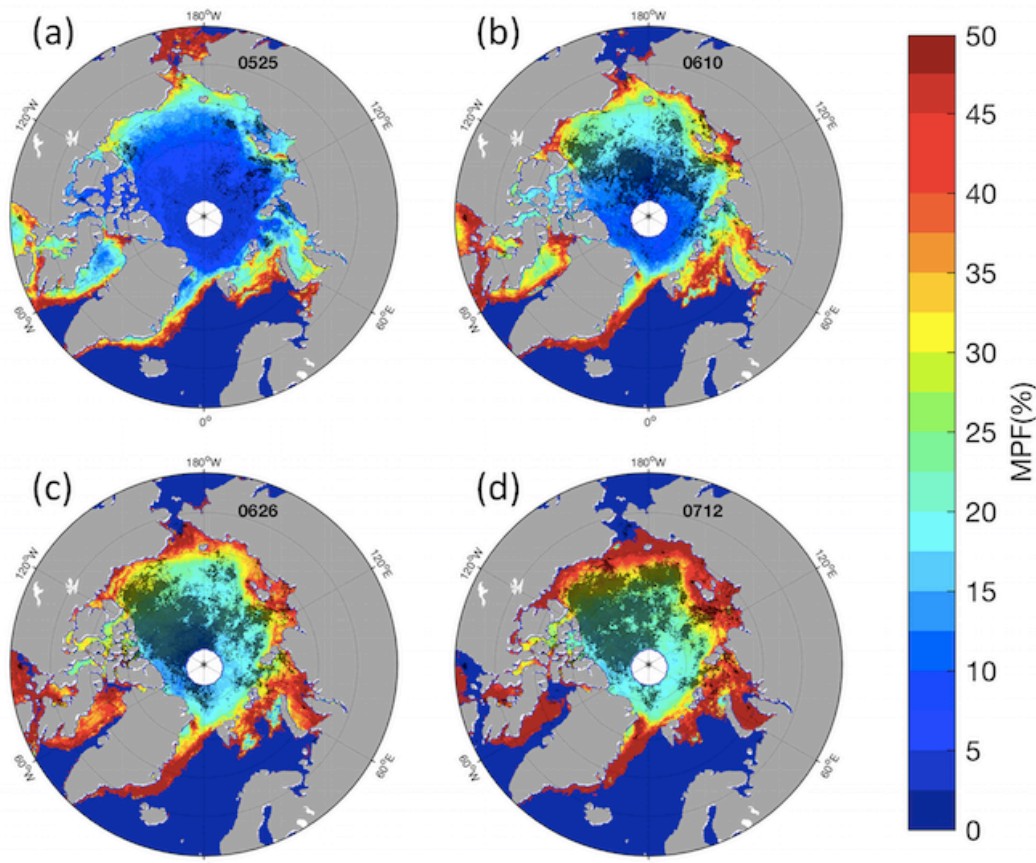

**Figure 12:** Distribution of the MPF from 9 May to the day given (a) 25 May, (b) 10 June, (c) 26 June, and (d) 12 July. Color is the averaged MPF relative to ice-covered area for the day given during 2000–2017. The dark dots (> 95% confidence level) are the statistically significant correlations between the MPF integrated from 9 May to the day given and September Arctic sea ice extent.





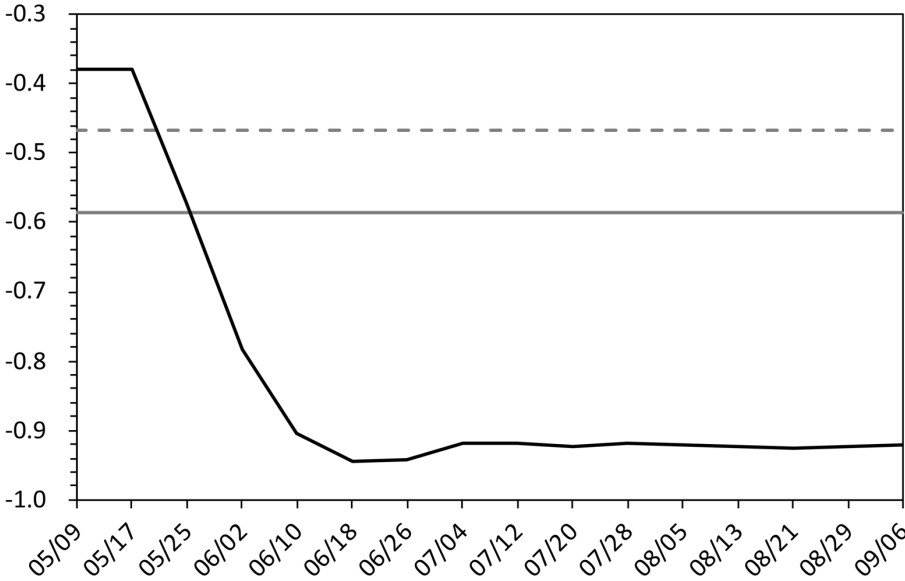

**Figure 13:** Correlation between time series of the MPF relative to ice-covered area (integrated from 9 May to the day given) and September sea ice extent. The horizontal grey lines are the 95% (dashed) and 99% (solid) confidence levels.




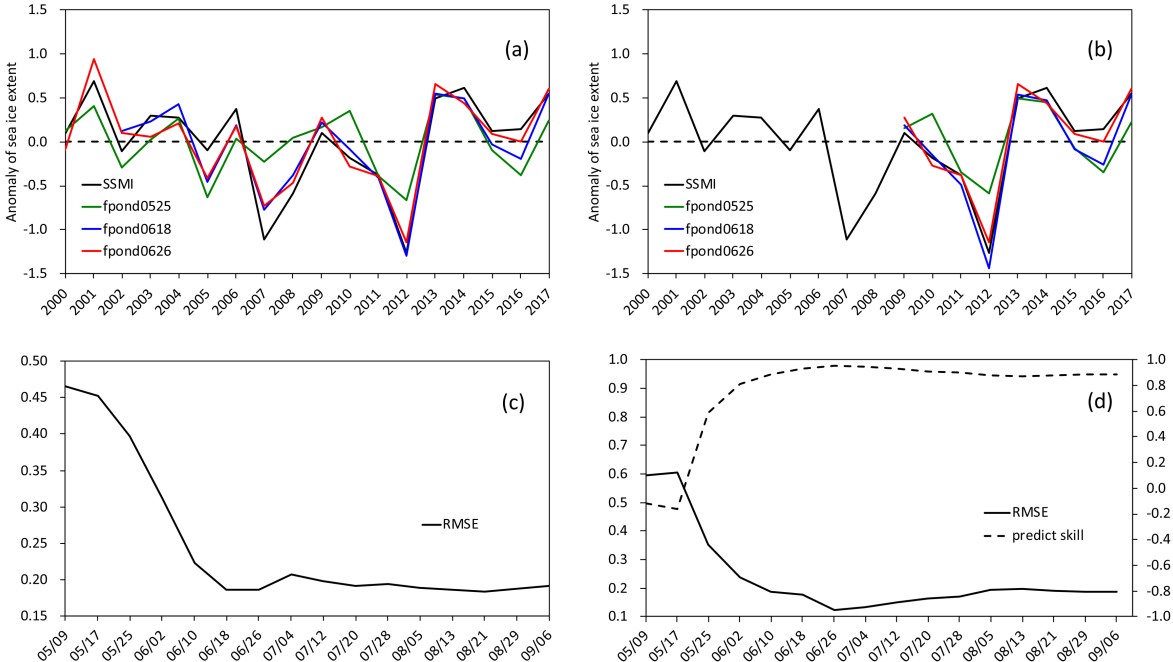

**Figure 14:** Regressed and predicted Arctic September sea ice extent anomaly (detrended). (a) Regressed ice extent anomalies for three different integration periods (9 May-25 May, 9 May-18 June, and 9 May-26 June) and (c) their regression errors. (b) Predicted ice extent anomalies and (d) their predicted errors (left y-axis) and forecast skills (right y-axis).






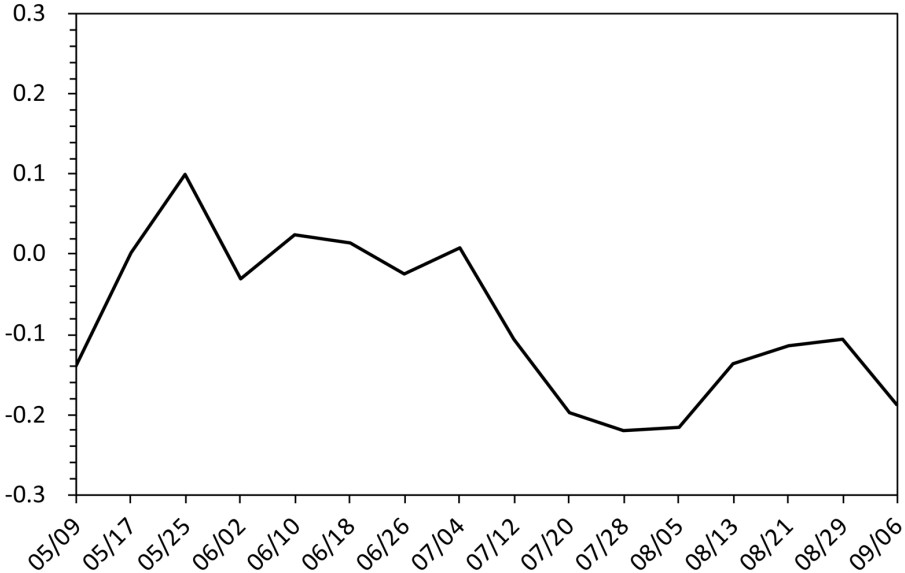

**Figure 15:** Correlation between time series of the cloud fraction (integrated from 9 May to the day given) and September sea ice extent.