# Peer review of "Investigation of spatiotemporal variability of melt pond fraction and its relationship with sea ice extent during 2000-2017 using a new data"

_The Cryosphere, 2019_

## Referee Comment (RC1) · Anonymous Referee #1 · 17 Sep 2019

The manuscript "Investigation of spatiotemporal variability of melt pond fraction and its relationship with sea ice extent during 2000-2017 using a new data" by Ding et al. submitted to The Cryosphere is on an important subject relevant for the journal. It is well written and the methods used seem to be adequate. However, it has a serious methodological flaw which needs at least a major revision or better more work and a re-submission.

The authors train a neural network with MODIS data and in-situ melt pond observations to retrieve the melt pond fraction (MPF). This is similar to the approach of Rösel et al. (2012) but with a major difference. Rösel et al. (2012) use the mixing equation to solve

for three unknown surface types: open water, melt ponds, and snow and ice. This means Rösel et al. estimate the melt pond fraction with respect to the ice surface. The sea ice concentration results as an independent quantity from the MODIS retrieval. In my understanding, the authors of the present manuscript do not retrieve the ice concentration as an independent parameter which means that the coverage of melt ponds is not correctly estimated in areas with ice concentration below 100%. This is obvious in gradients of the MPF in the marginal ice zone where a coverage of >50% is estimated (e.g. Fig. 11 and 12) . This is a clear artefact and does not resemble the real melt pond coverage. The new MPF seems to be highly influenced by the ice concentration and is not an independent measure, see Kern et al. (2016) for further details.

Kern, S., Rösel, A., Pedersen, L. T., Ivanova, N., Saldo, R., and Tonboe, R. T.: The impact of melt ponds on summertime microwave brightness temperatures and sea-ice concentrations, The Cryosphere, 10, 2217–2239, https://doi.org/10.5194/tc-10-2217-2016, 2016.

---

## Author Comment (AC1) · 24 Sep 2019

**Preliminary response to the review of TC-2019-208 "Investigation of spatiotemporal variability of melt pond fraction and its relationship with sea ice extent during 2000–2017 using a new data" by Yifan Ding, Xiao Cheng, Jiping Liu, Fengming Hui, and Zhenzhan Wang**

*General comments:*

"*The authors train a neural network with MODIS data and in-situ melt pond observations to retrieve the melt pond fraction (MPF). This is similar to the approach of Rösel et al. (2012) but with a major difference. Rösel et al. (2012) use the mixing equation to solve for three unknown surface types: open water, melt ponds, and snow and ice. This means Rösel et al. estimate the melt pond fraction with respect to the ice surface. The sea ice concentration results as an independent quantity from the MODIS retrieval. In my understanding, the authors of the present manuscript do not retrieve the ice concentration as an independent parameter which means that the coverage of melt ponds is not correctly estimated in areas with ice concentration below 100%. This is obvious in gradients of the MPF in the marginal ice zone where a coverage of >50% is estimated (e.g. Fig. 11 and 12). This is a clear artefact and does not resemble the real melt pond coverage. The new MPF seems to be highly influenced by the ice concentration and is not an independent measure, see Kern et al. (2016) for further details.*

*Kern, S., Rösel, A., Pedersen, L. T., Ivanova, N., Saldo, R., and Tonboe, R. T.: The impact of melt ponds on summertime microwave brightness temperatures and sea-ice concentrations, The Cryosphere, 10, 2217–2239, https://doi.org/10.5194/tc-10-2217-2016, 2016.*"

**Response:**

We appreciate the reviewer for the helpful comments on the manuscript. Our preliminary response is attached as a supplement. It should be noted that, we are currently working on the reviewer's comments (see the third point below). Therefore, this response only clarifies some issues in the comments. Later, we will upload another supplement with detailed results to respond to all the issues in the comments.

First, we would like to provide a description of the deep neural network training in the manuscript. For the training, the input data is the spectral reflectance from four bands of MODIS (Moderate Resolution Imaging Spectroradiometer Terra MOD09A1 Version 6, https://lpdaac.usgs.gov/products/mod09a1v006/) on the 500 m polar stereographic grid. The training (target) data is the field observed melt pond fraction (MPF) relative to grid from six different sources (HOTRAX, DLUT, TransArc, PRIC-Lei, NSIDC, NPI). In the present network, the training (target) data does not include sea ice concentration (SIC) which has been pointed out as a possible issue in the reviewer's comment. However, the field observed SIC from the six sources has been used to transform the MPF relative to sea ice to the MPF relative to grid in the network training.

Second, the reviewer mentioned that "*the authors of the present manuscript do not retrieve the ice concentration as an independent parameter which means that the coverage of melt ponds is not correctly estimated in areas with ice concentration below 100%*" and "*This is obvious in gradients of the MPF in the marginal ice zone where a coverage of >50% is estimated (e.g. Fig. 11 and 12)*". **In our results (from Fig. 5 to Fig. 15 as well as Table 2), we have transformed the output of MPF relative to gird to the MPF relative to sea ice (see our figure captions)**. It should be noted that Fig.1 to Fig.4 and Table 1 in the manuscript are based on the MPF relative to grid. For the

transformation, we used the SIC from Nimbus-7 SMMR and DMSP SSM/I-SSMIS Passive Microwave Data developed by a revised NASA Team algorithm (NASA Team SIC, https://nsidc.org/data/nsidc-0051). The NASA Team SIC data is independent from the MPF retrieved by our network. Therefore, all our analyses (from Fig. 5 to Fig. 15 as well as Table 2) are based on the MPF relative to sea ice, which means the MPF is estimated in areas with ice concentration below 100% (Note: we only consider the grid cell with NASA Team SIC greater than 15%, as mentioned in line 201 in the manuscript.). The grid cell with MPF greater than NASA Team SIC have been removed in our analysis. Table 1 (below) shows the percentage of grid cell with MPF greater than NASA Team SIC (Note: these grid cells are considered as bad retrieval). The results show that only less than 2% and less than 0.1% of the grid cells have bad MPF retrieval when considering grid cell with SIC>15% and SIC>30%, respectively. This means that the bad MPF retrievals are primarily located in the sea ice edge area (with small concentration).

Third, the reviewer mentioned that the network should also include the SIC as an independent quantity (Note: the current manuscript used the NASA Team SIC as an independent quantity to restrain the grid cell with retrieved MPF over sea ice cover area and make sure the MPF is smaller than the ice concentration.). In order to further address the reviewer's concern, we are currently trying to add the SIC as the training data in the network. The work is underway, and detailed results will be given in the later supplement.

Table 1. The percentage (%) of the grid cell with MPF relative to grid greater than NASA Team SIC

| Year | MPF> NASA Team SIC (SIC>15%) | MPF> NASA Team SIC (SIC>30%) | Total girds (average per day) |
|---|---|---|---|
| 2000 | 1.92 | 0.09 | 49127 |
| 2001 | 1.77 | 0.12 | 45253 |
| 2002 | 2.13 | 0.13 | 47358 |
| 2003 | 2.37 | 0.12 | 48097 |
| 2004 | 1.93 | 0.10 | 47545 |
| 2005 | 2.24 | 0.14 | 45805 |
| 2006 | 1.99 | 0.14 | 45281 |
| 2007 | 2.53 | 0.08 | 42082 |
| 2008 | 2.09 | 0.09 | 43445 |
| 2009 | 1.92 | 0.08 | 44937 |
| 2010 | 2.07 | 0.09 | 42775 |
| 2011 | 2.31 | 0.07 | 41503 |
| 2012 | 2.22 | 0.09 | 39476 |
| 2013 | 1.28 | 0.06 | 43269 |
| 2014 | 1.76 | 0.07 | 43127 |
| 2015 | 1.54 | 0.03 | 41843 |
| 2016 | 2.04 | 0.09 | 40403 |
| 2017 | 1.38 | 0.04 | 41081 |
| Average | 1.97 | 0.09 | 44023 |

---

## Referee Comment (RC2) · Anonymous Referee #1 · 25 Sep 2019

Thank you very much for the prompt response which enables a real interactive discussion.

I am still concerned about the correlation of ice concentration and your MPF retrieval. You mention that the bad MPF retrievals are primarily located in the sea ice edge area (with small concentration). For me this sounds like a clear argument that the MODIS MPF retrievals are not independent of the ice concentration. The problem is that the NASA TEAM SIC which you use for correction is estimated on a much coarser resolution. Moreover, the NASA TEAM SIC is strongly influenced by the occurrence of melt ponds. Unfortunately, I can not see how you solved this problem of different

spatial scales and correlated measurements.

---

## Short Comment (SC1) · Ding, Y., et al. · 5 Nov 2019

Comments to

Investigation of spatiotemporal variability of melt pond fraction and its relationship with sea ice extent during 2000-2017 using a new data

by

Ding, Y., et al.

Dear authors,

with great interest I read your interesting paper about a new attempt to apply ANN to MODIS optical / NIR reflectances with the purpose to retrieve melt-pond fraction on sea ice in the Arctic.

I have the following concerns / questions.

I can understand very well that your ultimate goal is to (finally) derive a long melt-pond fraction time series for Arctic sea ice so that you can improve its geophysical interpretation. This is needed and very timely.

However, I find that both, the description of your methodology (see 1) and the description and amount of your evaluation activities (see 2) seems not yet to justify usage and exploitation of the data set the way presented in the current manuscript. I believe, if I were you, then I would kick out almost all figures / discussions dealing with the geophysical interpretation of this (so far) insufficiently evaluated data set because these interpretations could be very misleading. I would suggest to purely focus on the algorithm and results and their evaluation; this would be more than enough stuff to publish.

1) Using ANN requires optimal understanding and preparation of the input data and an accurate description of how you actually applied the ANN. In your case this applies to both the reflectivity data and the melt-pond observations. While you state which MODIS data set you use [which one was used by Rösel et al.?] it is not clear i) which collection this is based on (4,5,6?), and ii) how accurate the cloud masking indeed is for the high-latitude regions. So far I found limited evidence in the documentation of MODIS reflectance data that particularly over high latitudes (in contrast to lower latitudes) the uncertainties / biases due to clouds, cloud shadows and fog have been substantially improved. It would be good to better specify what is meant by "low view angle, absence of clouds [quality flag used?], cloud shadows and aerosol loading" so that other scientists could repeat your analysis. Please also note my comment to Figures 10 and 11.

[Figure]

Furthermore, I am concerned about your training data sets. Their description is very short and does neither sufficiently explain the different degrees of reliability between ship-based visual observations and air-borne observations nor does it comment on the accuracy of the data. What is called "resolutions" seems to be the coverage of one observation "footprint". The description lacks which additional data are used (sea-ice concentration from these ship- and air-borne observations) and it lacks to give examples which allows the reader to get an impression about the actual kind of data you are using. What seems in addition to be stated insufficiently detailed is how these data are pre-processed to be used in the ANN.

2) Your evaluation and presentation of the results appears to be very global. The only "true" kind of evaluation figure is Figure 4 and if I am not mistaken then there aren't any figures showing inter-comparisons of the actual melt-pond fraction for single 8-day periods with independent data. Wouldn't it therefore be a good idea to i) include map-based inter-comparisons between, e.g. the Istomina et al. data set or the Rösel et al. data set and your results, ideally these come along with scatterplots and/or histograms of the actual distribution of the melt pond fraction; ii) include overlays of airborne (and in-situ) data on both the input MODIS reflectance data and the resulting melt-pond fraction. iii) include an investigation about how melt progress is seen in your data set and how it is seen in the evaluation data sets used - if possible. iv) include a detailed description of how you co-located the different data sets. v) include a detailed description of the accuracy of both your results and the data used for training and evaluation. This could (and should) involve to include information about the sea-ice concentration in Fig. 10 and Fig. 11. One could doubt that your results are independent of the actual sea-ice concentration due to the dominating impact of any open water on the brightness temperatures used for the sea-ice concentration data set you used as sea-ice mask (which was one of the things avoided by Rösel et al. for good reason). It is also not clear to me (and seems not to be described in the methods section overwhelmingly detailed) how the reflectances of open water and melt ponds are unmixed efficiently enough to identify open water as open water and to not identify an actual melt pond as

a certain fraction of open water as well.

- You motivate (in Sect. 2.1) the inclusion of a fourth spectral band with the fact that by this action you are able to better discriminate property changes within the snow pack. While this might be an advantage for the early phase of melt (which you could have explained in more detail) it seems not to be clearly stated how this could improve melt-pond fraction retrieval at a later stage. I guess, one of the main suggestions for improvement in the Rösel et al. paper was motivated by the change in spectral characteristics over the course of the melt season resulting in a different spectral response of melt ponds on MYI compared to melt ponds on FYI. This is where I hoped that your paper would enhance the current state-of-the-art but I have difficulties to see this in the paper yet.

- Lines 120-122: You use a standard sea-ice concentration product as a sea-ice mask. While this is fine, several questions immediately pop up: i) what is meant by "revised NASA Team algorithm (Cavalieri et al., 1996)"? The year of the reference makes clear that it cannot be the enhanced NASA Team algorithm". ii) what are the specifications of this data set in terms of spatial and temporal resolution and how did you pre-process the data to match with the MODIS data? iii) passive microwave concentration have biases during summer as has been discussed, e.g., in Comiso and Kwok in 1996: "Surface and radiative characteristics of the summer Arctic sea ice cover from multi-sensor satellite observations" and in Kern et al. in 2016: "The impact of melt ponds on summertime microwave brightness temperatures and sea-ice concentrations". Doesn't using such sea-ice concentration data sets as sea-ice mask therefore require a more in-depth description of how you used the data and how the expected bias in sea-ice concentration influences your melt-pond retrieval?

- It appears to me that you did not yet adequately cite the MODIS melt-pond fraction data set of Rösel et al. (2012) which you are using in your overall comparison (e.g. Figure 4). Would you mind to check which version of this data set you used and provide the doi and version of it in your reference list? I guess this would help other potential

users to locate the correct data set.

- Lines 148-153: I checked the Webster et al. [2015] paper. I have serious doubts that this is the correct reference. I found that this paper basically compares a new method to derive melt-pond fraction based on APLIS campaign data and compared the results with SHEBA data. I did not find the mentioned 2000-2014 MPF data set. Here you would appreciate a hint about where to find this potentially very valuable data set.

- Figure 5: This figure states an average (2000-2017) pan-Arctic melt-pond fraction of 10% already in the middle of May. This appears to be too large. While Liu et al. (2015) found a similar evolution they used the old Rösel et al. melt-pond fraction data set which was erroneously high and which has been corrected based on the findings presented in Mäkynen et al. (2014). As you state yourself in the paper, melt onset typically occurs early June and I'd even state the melt onset for the majority of the MYI is in late June / early July which is when you suggest a melt-pond fraction over MYI of 15% already.

- Figures 10 and 11: These are 8-daily estimates of the melt-pond fraction. How come that compared to the Rösel et al. product there are no gaps due to clouds? It appears to be very unlikely that the more recent collection of MODIS data you used does contain less pixels flagged as cloud covered. This gets back to my general concern about the degree of detail in the description of the input data and then also in the results.

---

## Referee Comment (RC3) · Anonymous Referee #2 · 18 Dec 2019

The manuscript is dedicated to the retrieval of melt pond fraction using remote sensing data. As input data, a higher level MODIS product (surface reflectance at 4 spectral channels) is used together with a set of training in situ data from various sources. The connection between the ground truth data and the observed surface reflectance is established by means of a multi-layer neural network. The obtained dataset is further compared to the existing melt pond fraction product and to the ice extent data, in order to check the prediction skill of the melt pond data in spring relative to the minimum ice extent in autumn, according to the well-known publication of Schroeder et al.The motivation for the presented work is solid and is well-presented. At the time of writing, there are no published long-term melt pond fraction data sets. The existing MODIS

data set (Rösel et al) is not continued, as well as the alternative product from MERIS (Zege et al, stopped due to sensor failure). Therefore, the topic of the publication is certainly up-to-date and a new melt pond fraction data set is of high importance to the scientific community. The topic of the manuscript fits into the scope of the journal as well.

However, the transparency and quality of the presentation is sometimes so poor that it is hard for the reviewer to decipher e.g. which data were used or which method was applied. The main concerns are listed below:

a) The weak points of the manuscript are the network training and the validation. It is far too early to include MPF trend and MPF map analysis before these are sorted out, as well as claim to outperform another retrieval. The provided description of the in situ training data, validation data, and validation results are insufficient and do not allow to assess the performance of the retrieval.

Please provide:

- a detailed description of the training and the validation datasets you use - current description is confusing and hard to understand. For each dataset, the size of the sample, spatial resolution, spatial coverage, temporal coverage, and the method of spatial and temporal collocation should be clearly stated. For each 8-day MODIS composite that you compare, how many days offset to in situ data do you allow? when you do that, do you have any assumptions about the evolution of MPF? how to do you train a neural network for 8-day composites using single day in situ data, and how do you compare those for validation? In a 8-day composite, which is not an 8-day average, you do not really know which day a given pixel stems from - or did you use this information?

- The Webster validation dataset which is your only independent validation dataset is either a typo or just wrong, there is no such dataset in that paper. Please double check.

Line 149-152: You state that the validation data by Webster et al 2015 supposedly

stems from 2000-2014 and has resolution 8 to 25km2. When I look into that manuscript I discover that the study by Webster et al 2015 is a fine approach to classify optical GFL images of 1m resolution, using collocated APLIS 2011 field campaign data - please see Table 1 in Webster et al 2015 for a list of the used data. These data are from 2011 only and have spatial resolution of 1 meter. I cannot find any other data in the manuscript by Webster et al 2015 which stems from 2000-2014 and has resolution 8-25km2.

- a scatter plot with "original MPF" and "retrieved MPF" on the axes, where each data point of the training and validation datasets can be seen, as well as the size of the sample, also for the Webster dataset. Your Fig. 4 cannot be used as the validation plot.

- make sure to use the original MODIS resolution and the finest spatial resolution of in situ data, both datasets also temporally collocated, to ensure a good quality of the comparison. For the transparency, it would be a good idea to provide case studies where you plot e.g. reference aerial values on your retrieved MODIS MPF map and dicsuss the discrepancies.

b) It is not sufficient to train a neural network only for melt ponds disregarding both open water and surface variability - it has been already mentioned by other reviewers and I 100% support this important concern. In the MPF maps (Fig. 10,11,12) the MPF along the ice edge stays constantly at the maximum value of 0.5 throughout the summer, although the FYI cannot hold the maximum pond fraction after melt peak due to the increased ice permeability and pond drainage (Polashenski thesis and other works). From this one can conclude that this high MPF value is rather connected to the low ice concentration at the ice edge and not to the MPF. Certainly, this problem is present not only at the ice edge, just not as clearly visible as at the ice edge. This issue is currently not solved, not discussed and has to be in some way addressed.

c) the structure of the manuscript: should you consider extending the descriptions, discussion, adding new plots and case studies as suggested, then the material from

3.2 onward would be far too much for one publication. You might also need to retrain the neural network for satellite from single days or include ice concentration in the equation, so the trends and MPF maps need to be updated as well. The reviewer suggests that you rethink and reduce the structure and focus on the quality of the research and the methodology first, so that the results that you claim would be clearly supported by your investigations.

---

## Author Comment (AC2) · 8 Mar 2020

**Response to the reviews of TC-2019-208 "Investigation of spatiotemporal variability of melt pond fraction and its relationship with sea ice extent during 2000–2017 using a new data" by Yifan Ding, Xiao Cheng, Jiping Liu, Fengming Hui, and Zhenzhan Wang**

We greatly appreciate the thoughtful comments from the reviewer. According to the reviewer's comments, we revised the original manuscript. All issues raised have been considered thoroughly.

*Round 1: General comment by reviewer #1*

"*The authors train a neural network with MODIS data and in-situ melt pond observations to retrieve the melt pond fraction (MPF). This is similar to the approach of Rösel et al. (2012) but with a major difference. Rösel et al. (2012) use the mixing equation to solve for three unknown surface types: open water, melt ponds, and snow and ice. This means Rösel et al. estimate the melt pond fraction with respect to the ice surface. The sea ice concentration results as an independent quantity from the MODIS retrieval. In my understanding, the authors of the present manuscript do not retrieve the ice concentration as an independent parameter which means that the coverage of melt ponds is not correctly estimated in areas with ice concentration below 100%. This is obvious in gradients of the MPF in the marginal ice zone where a coverage of >50% is estimated (e.g. Fig. 11 and 12). This is a clear artefact and does not resemble the real melt pond coverage. The new MPF seems to be highly influenced by the ice concentration and is not an independent measure, see Kern et al. (2016) for further details.*

*Kern, S., Rösel, A., Pedersen, L. T., Ivanova, N., Saldo, R., and Tonboe, R. T.: The impact of melt ponds on summertime microwave brightness temperatures and sea-ice concentrations, The Cryosphere, 10, 2217–2239, https://doi.org/10.5194/tc-10-2217-2016, 2016.*"

**Response:**

   First, we would like to provide a description of the deep neural network training in the manuscript. For the training, the input data is the spectral reflectance from four bands of MODIS (Moderate Resolution Imaging Spectroradiometer Terra MOD09A1 Version 6, https://lpdaac.usgs.gov/products/mod09a1v006/) on the 500 m polar stereographic grid. The training (target) data is the field observed melt pond fraction (MPF) relative to grid from six different sources (HOTRAX, DLUT, TransArc, PRIC-Lei, NSIDC, NPI). In the present network, the training (target) data does not include sea ice concentration (SIC) which has been pointed out as a possible issue in the reviewer's comment. However, the field observed SIC from the six sources has been used to transform the MPF relative to sea ice to the MPF relative to grid in the network training.

   Second, the reviewer mentioned that "the authors of the present manuscript do not retrieve the ice concentration as an independent parameter which means that the coverage of melt ponds is not correctly estimated in areas with ice concentration below

100%" and "This is obvious in gradients of the MPF in the marginal ice zone where a coverage of >50% is estimated (e.g. Fig. 11 and 12)". In our results (from Fig. 5 to Fig. 15 as well as Table 2 in the TCD manuscript), we have transformed the output of MPF relative to gird to the MPF relative to sea ice (see our figure captions). It should be noted that Fig.1 to Fig.4 and Table 1 in the TCD manuscript are based on the MPF relative to grid. For the transformation, we used the SIC from Nimbus-7 SMMR and DMSP SSM/I-SSMIS Passive Microwave Data developed by a revised NASA Team algorithm (NASA Team SIC, https://nsidc.org/data/nsidc-0051). The NASA Team SIC data is independent from the MPF retrieved by our network. Therefore, all our analyses (from Fig. 5 to Fig. 15 as well as Table 2 in the TCD manuscript) are based on the MPF relative to sea ice, which means the MPF is estimated in areas with ice concentration below 100% (Note: we only consider the grid cell with NASA Team SIC greater than 15%, as mentioned in line 201 in the manuscript.). The grid cell with MPF greater than NASA Team SIC have been removed in our analysis. Table 1 (below) shows the percentage of grid cell with MPF greater than NASA Team SIC (Note: these grid cells are considered as bad retrieval). The results show that only less than 2% and less than 0.1% of the grid cells have bad MPF retrieval when considering grid cell with SIC>15% and SIC>30%, respectively. This means that the bad MPF retrievals are primarily located in the sea ice edge area (with small concentration).

**Table 1.** The percentage (%) of the grid cell with MPF relative to grid greater than NASA Team SIC

| Year | MPF> NASA Team SIC (SIC>15%) | MPF> NASA Team SIC (SIC>30%) | Total girds (average per day) |
|---|---|---|---|
| 2000 | 1.92 | 0.09 | 49127 |
| 2001 | 1.77 | 0.12 | 45253 |
| 2002 | 2.13 | 0.13 | 47358 |
| 2003 | 2.37 | 0.12 | 48097 |
| 2004 | 1.93 | 0.10 | 47545 |
| 2005 | 2.24 | 0.14 | 45805 |
| 2006 | 1.99 | 0.14 | 45281 |
| 2007 | 2.53 | 0.08 | 42082 |
| 2008 | 2.09 | 0.09 | 43445 |
| 2009 | 1.92 | 0.08 | 44937 |
| 2010 | 2.07 | 0.09 | 42775 |
| 2011 | 2.31 | 0.07 | 41503 |
| 2012 | 2.22 | 0.09 | 39476 |
| 2013 | 1.28 | 0.06 | 43269 |
| 2014 | 1.76 | 0.07 | 43127 |
| 2015 | 1.54 | 0.03 | 41843 |
| 2016 | 2.04 | 0.09 | 40403 |
| 2017 | 1.38 | 0.04 | 41081 |
| Average | 1.97 | 0.09 | 44023 |

Third, the reviewer mentioned that the network should also include the SIC as an independent quantity (Note: the current manuscript used the NASA Team SIC as an independent quantity to restrain the grid cell with retrieved MPF over sea ice cover area and make sure the MPF is smaller than the ice concentration.). **In order to further address the reviewer's concern, we have re-trained the networks by adding the sea ice concentration (SIC) as the training (target) data. The results are shown below.**

**Round 2: Additional comment by reviewer#1**

"*I am still concerned about the correlation of ice concentration and your MPF retrieval. You mention that the bad MPF retrievals are primarily located in the sea ice edge area (with small concentration). For me this sounds like a clear argument that the MODIS MPF retrievals are not independent of the ice concentration. The problem is that the NASA TEAM SIC which you use for correction is estimated on a much coarser resolution. Moreover, the NASA TEAM SIC is strongly influenced by the occurrence of melt ponds. I cannot see how you solved this problem of different spatial scales and correlated measurements*"

**Response:**

To address the reviewer's concern, here we added observed SIC as the target data in the network training, and also retrieved SIC as the second output. We used the observed SIC from three independent sources as the target and trained the network separately. (note: the first output is MPF, the same as described in section 2 of TCD manuscript). Table 2 provides the detailed information.

**Table 2**. Details of the target and output for the network

| Network | Training Input | Training | Output (target) |
|---|---|---|---|
| DNN_MPF (no SIC) | | Observed MPF | MPF (no SIC) |
| DNN_MPF+NASASIC | MOD09A1 bands (Band 1, 2, 3, 5) | Observed MPF & NASA Team SIC | MPF + SIC |
| DNN_MPF+FieldSIC | | Observed MPF & Observed SIC | |
| DNN_MPF+AMSRSIC | | Observed MPF & AMSR-SIC | |

- DNN_MPF (no SIC) is the network trained in the TCD manuscript. The training input is the four MOD09A1 bands (Band 1, 2, 3, 5) on the 500 m polar stereographic grid. The training output is the observed MPF from six sources (HOTRAX, DLUT, TransArc, PRIC-Lei, NSIDC, NPI). The DNN_MPF (no SIC) does not include SIC as the target in the network training.

- DNN_MPF+NASASIC is the network trained by adding the NASA Team SIC (Cavalieri et al., 1996) as the second target. The NASA Team SIC is derived from

Nimbus-7 SMMR and DMSP SSM/I-SSMIS Passive Microwave Data using a revised NASA Team algorithm (https://nsidc.org/data/nsidc-0051). In the network training, the NASA Team SIC was resampled from 25 km to the 500 m polar stereographic grid to match the resolution of the MODIS surface reflectance.

• DNN_MPF+FieldSIC is the network trained by adding the observed SIC from multi-sources (HOTRAX, DLUT, TransArc, PRIC-Lei, NSIDC and NPI) as the second target. The observed SIC is obtained from the same sources as the observed MPF. In the network training, the observed SIC was resampled from its original resolution (coverage) to the 500 m polar stereographic grid to match the resolution of MODIS surface reflectance (note: we use the average of the observed SIC from each source located in the same grid as the resampled SIC).

• DNN_MPF+AMSRSIC is the network trained by adding the SIC derived from Advanced Microwave Scanning Radiometer-Earth Observing System and Advanced Microwave Scanning Radiometer 2 (hereafter referred to as AMSR SIC, Spreen et al., 2008) as the second target. The AMSR SIC is developed by the University of Bremen using the ARTIST Sea Ice (ASI) algorithm (https://seaice.uni-bremen.de/sea-ice-concentration). In the network training, the AMSR SIC was resampled from 6.25 km to the 500 m polar stereographic grid to match the resolution of MODIS surface reflectance.

For the final MPF and SIC data retrieval, the data on the 12.5 km polar stereographic grid were used in the ensemble-based network (note: MOD09A1 on the 12.5 km polar stereographic grid was used as the input). The only difference between DNN_MPF (no SIC) and the other three networks (DNN_MPF+NASASIC, DNN_MPF+FieldSIC and DNN_MPF+AMSRSIC) is that the three networks contain SIC as the second target in network training. Therefore, the final dataset from DNN_MPF (no SIC) only contains MPF on the 12.5 km polar stereographic grid and the final dataset from the other three networks contains MPF and SIC on the 12.5 km polar stereographic grid.

Figure 1 shows the correlation coefficients and the RMSE of MPF from the above four network training. It appears that the correlation coefficients of the four networks with independent SIC are comparable. This is also true for the RMSE. This suggests that the influence of the ice concentration on the retrieved MPF is minor. This further increases the reliability of our MPF retrieval. We check the spatial correlation coefficients and RMSE of the MPF from three re-trained networks with the MPF from DNN_MPF (no SIC) in each year during 2000-2017. The results show that the average spatial correlation coefficient is ~0.99 and the RMSE is ~0.012. This suggests that the MPF from the re-trained networks are generally consistent with that from DNN_MPF (no SIC).

[Figure]

**Figure 1.** Validation of the MPF from four networks against the observed MPF: (a) correlation coefficients and (b) RMSE. (repetition of Fig.4 in the TCD manuscript).

For further comparison, we show the MPF (relative to grid) in 2017 from DNN_MPF (no SIC) and the three re-trained networks (DNN_MPF+NASASIC, DNN_MPF+FieldSIC and DNN_MPF+AMSRSIC). The results show that the spatial MPF during May to September in 2017 from DNN_MPF (no SIC) (Fig.2) are almost the same with that from the three networks added SIC (Fig.3 to 5). This further suggests that the SIC only has very limited effect on the MPF retrieval in our method.

[Figure]

**Figure 2.** The evolution of the MPF from DNN_MPF (no SIC) relative to grid from early May to early September in 2017.

[Figure]

**Figure 3.** Same as Fig.2, except for the MPF from DNN_MPF+NASASIC.

[Figure]

**Figure 4.** Same as Fig.2, except for the MPF from DNN_MPF+FieldSIC.

[Figure]

**Figure 5.** Same as Fig.2, except for the MPF from DNN_MPF+AMSRSIC.

Table 3 shows the percentage of grid cell with MPF greater than SIC (regarded as bad retrieval). The MPF (relative to grid) and SIC used here are both from the three re-trained networks (DNN_MPF+NASASIC, DNN_MPF+FieldSIC and DNN_MPF+AMSRSIC). The results show that 0.84-1.31% of the grid cells have bad MPF retrieval when considering grid cell with SIC>15%. It can be reduced to 0.05-0.19% of the grid cells when considering SIC>30%. The bad retrieval (MPF larger than SIC) has been removed in the analyses. Compared to Table 1 in the preliminary response to the review#1, the percentage of the grid with MPF larger than SIC does not change much whether the MPF is from DNN_MPF (no SIC) or the three re-trained networks (note: 1.97% and 0.09% of the grid cells have bad MPF retrieval when considering grid cell with SIC>15% and SIC>30% in DNN_MPF (no SIC)). This suggests that the SIC has very limited effect on the MPF retrieval in our method, which further increases the reliability of our method.

In order to minimize the bad MPF retrievals that are primarily located in the sea ice edge area with small concentration. In this revision, we only consider the grid cell with sea ice concentration greater than 30%, instead of 15%. The original MPF from DNN_MPF (no SIC) has been replaced by the retrieval from DNN_MPF+NASASIC.

**Table 3.** The percentage of the grid cell with MPF relative to grid greater than SIC

| Year | MPF > Retrieved SIC | | | | | | Total grids |
| | DNN_MPF+NASASIC | | DNN_MPF+FieldSIC | | DNN_MPF+AMSRSIC | | |
| | SIC>15% | SIC>30% | SIC>15% | SIC>30% | SIC>15% | SIC>30% | |
|---|---|---|---|---|---|---|---|
| 2000 | 1.85 | 0.17 | 1.29 | 0.08 | 1.14 | 0.27 | 49127 |
| 2001 | 1.45 | 0.13 | 0.92 | 0.03 | 0.77 | 0.22 | 45253 |
| 2002 | 1.30 | 0.10 | 1.02 | 0.04 | 0.84 | 0.21 | 47358 |
| 2003 | 1.50 | 0.13 | 1.19 | 0.07 | 1.01 | 0.18 | 48097 |
| 2004 | 1.29 | 0.12 | 0.96 | 0.04 | 0.96 | 0.20 | 47545 |
| 2005 | 1.46 | 0.12 | 1.18 | 0.06 | 1.00 | 0.22 | 45805 |
| 2006 | 1.60 | 0.12 | 1.21 | 0.06 | 1.05 | 0.25 | 45281 |
| 2007 | 1.49 | 0.11 | 1.21 | 0.05 | 1.04 | 0.20 | 42082 |
| 2008 | 1.52 | 0.11 | 1.41 | 0.09 | 1.18 | 0.16 | 43445 |
| 2009 | 1.71 | 0.13 | 1.44 | 0.09 | 1.24 | 0.22 | 44937 |
| 2010 | 1.50 | 0.12 | 1.17 | 0.04 | 0.86 | 0.22 | 42775 |
| 2011 | 1.42 | 0.10 | 1.21 | 0.05 | 0.86 | 0.19 | 41503 |
| 2012 | 0.93 | 0.09 | 0.91 | 0.03 | 0.57 | 0.11 | 39476 |
| 2013 | 1.05 | 0.08 | 0.85 | 0.02 | 0.44 | 0.13 | 43269 |
| 2014 | 1.23 | 0.09 | 1.05 | 0.05 | 0.99 | 0.17 | 43127 |
| 2015 | 0.66 | 0.07 | 0.57 | 0.01 | 0.33 | 0.11 | 41843 |
| 2016 | 0.87 | 0.08 | 0.62 | 0.02 | 0.42 | 0.16 | 40403 |
| 2017 | 0.82 | 0.08 | 0.78 | 0.05 | 0.49 | 0.11 | 41081 |
| Average | 1.31 | 0.11 | 1.06 | 0.05 | 0.84 | 0.19 | 44023 |

Reference:

Cavalieri, D. J., Parkinson, C. L., Gloersen, P., and Zwally, H. J.: Sea Ice Concentrations from Nimbus-7 SMMR and DMSP SSM/I-SSMIS Passive Microwave Data, Version 1 (updated yearly), NASA National Snow and Ice Data Center Distributed Active Archive Center, https://doi.org/10.5067/8GQ8LZQVL0VL, 1996.

Spreen, G., Kaleschke, L., and Heygster, G.: Sea ice remote sensing using AMSR-E 89 GHz channels, J. Geophys. Res., 113, C02S03, https://doi.org/10.1029/2005JC003384, 2008.

---

## Author Comment (AC3) · 8 Mar 2020

**Response to the short comments of TC-2019-208 "Investigation of spatiotemporal variability of melt pond fraction and its relationship with sea ice extent during 2000–2017 using a new data" by Yifan Ding, Xiao Cheng, Jiping Liu, Fengming Hui, and Zhenzhan Wang**

We greatly appreciate the thoughtful comments from Dr. Stefan Kern. According to the comments, we revised the original manuscript. All issues raised have been considered thoroughly.

**Comments by Dr. Stefan Kern**

"*1) Using ANN requires optimal understanding and preparation of the input data and an accurate description of how you actually applied the ANN. In your case this applies to both the reflectivity data and the melt-pond observations. While you state which MODIS data set you use [which one was used by Rösel et al.?] it is not clear i) which collection this is based on (4,5,6?), and ii) how accurate the cloud masking indeed is for the high-latitude regions. So far I found limited evidence in the documentation of MODIS reflectance data that particularly over high latitudes (in contrast to lower latitudes) the uncertainties / biases due to clouds, cloud shadows and fog have been substantially improved. It would be good to better specify what is meant by "low view angle, absence of clouds [quality flag used?], cloud shadows and aerosol loading" so that other scientists could repeat your analysis. Please also note my comment to Figures 10 and 11. Furthermore, I am concerned about your training data sets. Their description is very short and does neither sufficiently explain the different degrees of reliability between ship-based visual observations and air-borne observations nor does it comment on the accuracy of the data. What is called "resolutions" seems to be the coverage of one observation "footprint". The description lacks which additional data are used (sea ice concentration from these ship- and air-borne observations) and it lacks to give examples which allows the reader to get an impression about the actual kind of data you are using. What seems in addition to be stated insufficiently detailed is how these data are pre-processed to be used in the ANN.*"

**Response:**

Based on the reviewer's comments, in this revision, we added more detailed descriptions about data and methods used in the manuscript.

**1)** We provided more information about the MODIS data used in this study, which is the MODIS/Terra Surface Reflectance 8-Day L3 Global 500m SIN Grid V006 (MOD09A1 version 6, https://lpdaac.usgs.gov/products/mod09a1v006/, Vermote, 2015). Note that the MODIS data used in Rösel et al. (2015) is the MOD09A1 version 5. Four spectral bands of MOD09A1 version 6 were used in our study (as the input data to the deep neural network, see **section 4)** for details) to derive MPF, including band 1, bandwidth of 620-670 nm; band 2, bandwidth of 841-876 nm; band 3, bandwidth of 459-479 nm; band 5, bandwidth of 1230-1250 nm. The improvements of MOD09A1 version 6 include: "a) Improvements to the aerosol retrieval and correction algorithm along with new aerosol retrieval look-up tables; b) Refinements to the internal snow,

cloud, and cloud shadow detection algorithms. Uses Bidirectional Reflectance Distribution Function (BRDF) database to better constrain the different threshold used; c) Processes ocean bands to create a new Surface Reflectance Ocean product and provides Quality Assurance (QA) datasets for these bands; d) Improved discrimination of salt pans from cloud and snow, along with the inclusion of a salt pan flag in the QA band." (https://lpdaac.usgs.gov/products/mod09a1v006/). The MOD09A1 version 6 has a spatial resolution of 500 m and is available at 8-day interval. "Each pixel contains the best possible L2G (the Level 2G format, consisting of gridded Level 2 data, was developed as a means of separating geolocating from compositing and averaging) observation during an 8-day period as selected on the basis of high observation coverage, low view angle, absence of clouds or cloud shadow, and aerosol loading." (MODIS Surface Reflectance User's Guide, https://lpdaac.usgs.gov/documents/306/MOD09_User_Guide_V6.pdf ). According to the MODIS Surface Reflectance User's Guide Collection 6, each orbit observation is assigned a score, based on whether it is flagged for cloud, cloud shadow, high aerosol or low aerosol, or contains high view angle or low solar zenith angle. The lowest score, 0, is assigned to observations with fill values for data. The remaining scores are:

1   BAD: data derived from a faulty or poorly corrected L1B pixel
2   HIGHVIEW: data with a high view angle (60 degrees or more)
3   LOWSUN: data with a high solar zenith angle (85 degrees or more)
4   CLOUDY: data flagged as cloudy or adjacent to cloud
5   SHADOW: data flagged as containing cloud shadow
6   UNCORRECTED: data flagged as uncorrected
7   CLIMAEROSOL: data flagged as containing the default level of aerosols
8   HIGHAEROSOL: data flagged as containing the highest level of aerosols
9   SNOW: data flagged as snow
10  GOOD: data which meets none of the above criteria

The observation with the highest score and the lowest view angle is selected for the MOD09A1, which minimizes the effect of the clouds on the spectral reflectance.

**2)** We provided more information about the in-situ data used in this study. The observed MPF relative to grid (or image area) from six different sources (HOTRAX, DLUT, TransArc, PRIC-Lei, NSIDC, and NPI) were used in our study (as the target data in the deep neural network, see **section 4)** for details).

- HOTRAX: MPFs were collected during the Healy Oden Trans-Arctic Expedition (HOTRAX) by the Polar Science Center, University of Washington (Perovich et al., 2009). The HOTRAX was conducted from August to September 2005 to obtain physical properties of the ice pack. The cruise started from Alaska, crossed the Bering, Chukchi, and Beaufort Seas and the Arctic Ocean reaching the North Pole, and then headed south and exited the Arctic basin through Fram Strait. The ice survey was made based on ice station measurements, helicopter survey flights, and the deployment of autonomous ice mass balance buoys. Fractional areas of melt

ponds were estimated during the expedition. The MPFs from HOTRAX used in the network training were measured at 77°-79°N and 84°-87°N on 13, 21, 29 August and 6 September 2005. The coverage of each MPF measurement is about 57 m×70 m. The obtained measurement from HOTRAX is the MPF relative to the grid (the coverage of each observation). The data can be found at http://psc.apl.uw.edu/data/.

- DLUT: MPFs were collected during two Chinese Arctic Research Expeditions by the Dalian University of Technology (DLUT, Lu et al., 2010; Huang et al., 2016). The first survey of DLUT was conducted from July to September 2008 during the third Chinese Arctic Research Expedition. During the cruise, eight helicopter flights were conducted and more than 9000 aerial images were obtained in the Pacific sector of the Arctic. The MPF was estimated from the digital image with a camera resolution of 3264×2248 pixels. The flight altitude generally varied around 100 m according to weather conditions. At this height, each snapshot covers an area of approximately 98 m×67 m (Lu et al., 2010). The second survey of DLUT was conducted from July to September 2010. The underway ship- and helicopter-based ice observations were primarily in the Chukchi Sea, Beaufort Sea, Canada Basin and Central Arctic Ocean. The images were classified into three distinct surface categories (sea ice/snow, water and melt ponds). The areal fraction of each category is determined by a camera resolution of 3264×2248 pixels. The flight altitude varied between 150 m to 500 m. Each image covers an area between 147 m×100 m and 490 m×335 m (Huang et al., 2016). The images from the two cruises are spaced without overlapping, and each image represents an independent scene. The DLUT MPF used in the network training was measured at 84°N and 86°N on 20 August 2008 and 13 August 2010. The coverage of each MPF estimated from the airborne image is about 98 m×67 m (first survey), 147 m×100 m and 490 m×335 m (second survey). The obtained measurement from DLUT is the MPF relative to the grid (image area).

- TransArc: MPFs were collected from the ice breaker RV Polarstern during the Germany Trans-Polar cruise ARKXXVI/3 (Nicolaus et al., 2012, hereafter referred to as TransArc). The TransArc conducted from August to October 2011. Visual observations of sea ice conditions were performed hourly from the bridge of Polarster. Sea ice type and thickness, snow depth, pond coverage, and surface scattering layer depth were recorded during the cruise. The observations followed the ASPeCT protocol with additional observations on melt ponds. It should be noted that the TransArc MPF was recorded on multiyear and first-year ice respectively for some cases, the MPF was estimated by using the linear mix of these values. The recording visibility in TransArc ranges between 50 m to 10 km based on ASPeCT (https://epic.awi.de/id/eprint/31658/14/ASPeCt_metcodes.pdf). The TransArc MPF used in the network training was measured at 84°-87°N on 13, 29 August and 6 September 2011 and the visibility of the records ranges from 500 m to 1000 m. The obtained measurement from TransArc is the MPF relative to sea ice cover. The data can be found at https://doi.pangaea.de/10.1594/PANGAEA.803312.

- PRIC-Lei: MPFs were collected during the Arctic Research Expeditions by the Polar Research Institute of China in summer from 2010 to 2016 (Lei et al., 2017, hereafter referred to as PRIC-Lei). Half-hourly Arctic Shipborne Sea Ice Standardization Tool (ASSIST) observations were conducted at the bridge of the R/V Xuelong to document sea ice concentration, sea ice and snow thickness, fractions of melt ponds (the area ratio relative to sea ice cover), dirty ice (with severe impurity depositions) and ridging, and floe size. Sea ice concentration was only assessed for a local area with a diameter of 2 km, which was reduced to 1 km on foggy days and melt pond fraction was estimated around the ship within 1 km. The MPF of PRIC-Lei used in the network training was measured at 73°-88°N on 28 July, 5 and 21 August 2010, 79-84°N on 12, 20, 28 August 2012, 73-79°N on 5, 13, 29 August 2014 and 73-80°N on 27 July, 4 and 20 August 2016. The coverage of each MPF record is about 1 km×1 km. The measurement from PRIC-Lei is MPF relative to sea ice cover.

- NSIDC: The MPFs were obtained from the NSIDC during summer of 2000 and 2001 (Fetterer et al., 2008). Development of this data set was based on experience gained using reconnaissance imagery during the Surface Heat Budget of the Arctic Ocean (SHEBA) and earlier summer ice monitoring experiments (NSIDC 2000, Fetterer and Untersteiner 1998). Visible band imagery from high-resolution satellites were acquired over the Beaufort Sea, the Canadian Arctic, the Fram Strait, and the East Siberian Sea during summer of 1999, 2000 and 2001. Imagery was analyzed using supervised maximum likelihood classification to derive either two (water and ice) or three (pond, open water, and ice) surface classes. The estimated pond coverage was under 500 m square cells within 10 km square images (image resolution is 1 m). The NSIDC MPF used in the network training was estimated from June to August in 2000 and 2001. The coverage of each MPF estimation is 500 m×500 m. The measurement from NSIDC is the MPF relative to grid (the coverage of each observation). The data can be found at https://nsidc.org/data/G02159/versions/1.

- NPI: The MPFs were collected by the Norwegian Polar Institute (NPI) during the field campaign on Arctic sea ice north of Svalbard in summer 2012 (Divine et al., 2015; Divine et al., 2016). The data set presents regional scale of about 150 km morphological properties of a relatively thin, 70-90 cm modal thickness, first-year Arctic sea ice pack in an advanced stage of melt. The data comprises fractions of three surface types (bare ice, melt ponds, and open water) along the flight tracks calculated from images acquired by a helicopter-borne camera system during ice-survey flights from late July to early August 2012. For a typical flight altitude of about 35 m over sea ice, the camera lenses used in the setup provide a footprint of about 60 by 40 m. For typical helicopter roll (pitch) angles of about −2° (1°), the distortion of the image plane from an ideal rectangular one and the associated uncertainty in the image area of less than 1% is considered insignificant. Therefore no correction for pitch and roll was applied to the images (Divine et al., 2015). The NPI MPF used in the network training was measured at 80-82°N on 3 August 2012. The coverage (footprint) of each MPF record is about 60 m×40 m. The obtained

measurement from NPI is the MPF relative to sea ice. The data can be found at https://data.npolar.no/dataset/5de6b1e4-b62f-4bd4-889c-8eb7bb862d3b.

Figures 1 to 6 show the observed MPF (used as target data in the network training) in the original resolution from above six sources overlaid on the NASA Team sea ice concentration (SIC). The MPF here is the fraction relative to the grid area. It appears that most of the MPF observations are in the grid with SIC above 40%.

[Figure]

**Figure 1.** Observed MPF from HOTRAX overlaid on NASA Team SIC.

[Figure]

**Figure 2.** Observed MPF from DLUT overlaid on NASA Team SIC.

[Figure]

**Figure 3.** Observed MPF from TransArc overlaid on NASA Team SIC.

[Figure]

**Figure 4.** Observed MPF from PRIC-Lei overlaid on NASA Team SIC.

[Figure]

**Figure 5.** Observed MPF from NSIDC overlaid on NASA Team SIC.

[Figure]

**Figure 6.** Observed MPF from NPI overlaid on NASA Team SIC.

**3)** We provided information from two additional in-situ observations, which are used as the completely independent validation data in this study (note: we add observations from JOIS as another completely independent validation dataset in this revision).

- Webster: The MPFs were retrieved by the Polar Science Center, University of Washington based on the classified high-resolution visible band satellite images following Webster et al. (2015). The image data source has been referred to as Global Fiducial Imagery, Literal Image Derived Products, National Technical Means images, and MEDEA Measurements of Earth Data for Environmental Analysis. The MPFs were measured at 69-86.5°N over the Beaufort Sea, Chukchi Sea, the Canadian Arctic, the Fram Strait, and the East Siberian Sea from May to August for the period of 1999-2014. The scene size (square grid) of the MPFs ranges from 5 to 25 km. The obtained measurement from Webster is the MPF relative to sea ice cover. In validation, the MPF has been transferred to the fraction relative to the grid (image area) using the measured SIC from Webster. The data and detailed description can be found at http://psc.apl.uw.edu/melt-pond-data/.

- JOIS: The MPFs were collected from the ship-based observations by Joint Ocean Ice Study (JOIS). The JOIS was conducted during 2003-2014 on the Canadian Coast Guard Ship Louis S. St-Laurent (Tanaka et al., 2016). The forward-looking camera imagery were gathered by two types of devices, a KADEC-EYE in 2005 and a Netcam-XL during 2008-2014. The cameras were mounted with a view of the horizon and ice pack in front of the ship. The images were classified into five types (water only; ice only; water and ice; pond and ice; water, pond and ice). Due to the camera malfunction and other bad ice conditions, information was missing in some years (Tanaka et al., 2016). The MPFs used here were obtained during the JOIS2011, measured at 68.5-88.5°N from 19 July to 11 September. The image in 1024×768 pixel was taken every 1-10 minute by Netcam-XL and the ice areas sampled per image range from 1453 to 2397 $m^2$. The total amount of the images is 34233. The obtained measurement from JOIS is the MPF relative to the grid (image area).

Figure 7 shows the observed MPF in the original resolution from JOIS overlaid on NASA Team SIC. Since the MPF from Webster is a single value on each observation date, we show the SIC of the observed MPF using scatter plot (Figure 8). Here the MPF is the fraction relative to the grid. The results show that most of the observations from JOIS are within the grids with SIC above 40%. The MPF from Webster are mainly measured at SIC above 60%.

[Figure]

**Figure 7.** Observed MPF from JOIS overlaid on NASA Team SIC.

[Figure]

**Figure 8.** Observed MPF from Webster against the Observed SIC.

**4)** Here we describe the way of the network training using the 8-day composite of MODIS surface reflectance and a specific day in-situ MPF measurement. For example, corresponding to a MPF observation from NSIDC on 4 July 2000 used as the training (target) data in the network, the surface reflectance from MOD09A1 (8-day composite) used as the input data in the network was obtained from the data file named "2000.07.03" (https://e4ftl01.cr.usgs.gov/MOLT/MOD09A1.006/). That means the date spanning of this MOD09A1 file is 3 July 2000 and 10 July 2000, which covers the MPF observation date. This is also applied to the validation.

For each 8-day composite of MOD09A1, we have 40 tiles (h09v02-h26v02, see https://modis-land.gsfc.nasa.gov/MODLAND_grid.html for details) in total to cover the entire Arctic. We mosaiced all the tiles into one *.hdf file using the MODIS Reprojection Tool (MRT) and then reprojected the mosaic to a GeoTIFF on the 500 m polar stereographic grid using ArcGIS. Each band (band 1, 2, 3 and 5) of the MOD09A1 was stored as a separate GeoTIFF file. For the network training, the input is the surface reflectance from the GeoTIFF files of the four bands on the 500 m polar stereographic grid.

For the observed MPFs from each source, we use the corresponding latitude and longitude to determine which gird cell (500 m polar stereographic grid) the observation falls in. If more than one observation from one source on a specific day fall in the same 500 m polar stereographic grid, the average of those observations is used as the training (target) data in the network. Note: the observed MPF relative to sea ice area has been transformed to the MPF relative to the grid (image area or coverage of each observation) based on the observed SIC in the network training.

In this study, we construct an ensemble-based deep neural network (hereafter referred to as DNN). The input of the network training is the four bands (band 1, 2, 3 and 5) of MOD09A1 on the 500 m polar stereographic grid. The training (target) data is the observed MPF relative to the grid (image area or coverage of each observation) from six sources (HOTRAX, DLUT, TransArc, PRIC-Lei, NSIDC, NPI). We choose the MOD09A1 from the file which covers the observation date as described above. It should be noted that in the network training, we only consider the grids that meet the following conditions: i) the values of MOD09A1 band 1, 2, 3, 5 are all within the valid range (MODIS Surface reflectance User guide collection 6, https://lpdaac.usgs.gov/documents/306/MOD09_User_Guide_V6.pdf); ii) the observed MPF is above 0 and below 100%; iii) the observed SIC (with MPF considered) relative to the gird is larger than the MPF relative to the grid.

For the final MPF data retrieval, the aforementioned GeoTIFF files were resampled from the 500 m to 12.5 km polar stereographic grid using the mean in a 25×25 window size by considering the valid data range of MOD09A1. We then apply the obtained DNN as mentioned above to derive the MPF dataset on the 12.5 km polar stereographic grid. The input for retrieving the MPF dataset are the four bands of MOD09A1 on the 12.5 km polar stereographic grid. The output is the MPF relative to the grid on the 12.5 km polar stereographic grid. For validation with the retrieved MPF on the 12.5 km polar stereographic grid, the average of the corresponding observations is calculated within the 12.5 km grid cell.

To further address the concern, we also trained the networks using the daily MODIS surface reflectance from MOD09GA (https://lpdaac.usgs.gov/products/mod09gav006/, Vermote and Wolfe, 2015), instead of the 8-day composite MOD09A1, on the 500 m polar stereographic grid and in-situ observations. The results are shown in **section 6)**.

"*2) Your evaluation and presentation of the results appears to be very global. The only "true" kind of evaluation figure is Figure 4 and if I am not mistaken then there aren't any figures showing inter-comparisons of the actual melt-pond fraction for single 8-day periods with independent data. Wouldn't it therefore be a good idea to i) include mapbased inter-comparisons between, e.g. the Istomina et al. data set or the Rösel et al. data set and your results, ideally these come along with scatterplots and/or histograms of the actual distribution of the melt pond fraction; ii) include overlays of airborne (and in-situ) data on both the input MODIS reflectance data and the resulting melt-pond fraction. iii) include an investigation about how melt progress is seen in your data set and how it is seen in the evaluation data sets used - if possible. iv) include a detailed description of how you co-located the different data sets. v) include a detailed description of the accuracy of both your results and the data used for training and evaluation. This could (and should) involve to include information about the sea-ice concentration in Fig. 10 and Fig. 11. One could doubt that your results are independent of the actual sea-ice concentration due to the dominating impact of any open water on the brightness temperatures used for the sea-ice concentration data set you used as sea-ice mask (which was one of the things avoided by Rösel et al. for good reason). It is also not clear to me (and seems not to be described in the methods section overwhelmingly detailed) how the reflectances of open water and melt ponds are unmixed efficiently enough to identify open water as open water and to not identify an actual melt pond as a certain fraction of open water as well.*"

**Response:**

**5)** We provided the map-based comparisons between our MPF data (hereafter referred to as DNN_MPF) and the MPF from Rösel et al. (2015) (https://icdc.cen.uni-hamburg.de/1/daten/cryosphere/arctic-meltponds.html, hereafter referred to as UH_MPFv2). The DNN_MPF is retrieved from DNN_MPF+NASASIC (see details in **section 12)**). Note: Both the DNN_MPF and UH_MPFv2 are retrieved from the 8-day composite of MODIS, but different version of MODIS (DNN_MPF from version 6 and UH_MPFv2 from version 5). Here we included one more MPF product. That is the MPF from Istomina et al. (2015) (https://seaice.uni-bremen.de/databrowser/#p=MERIS_fraction, hereafter referred to as UB_MPF). The UB_MPF consists of daily averages of the MPF retrieved from MERIS (Medium Resolution Imaging Spectrometer) swath Level 1b data using the MPD (Melt Pond Dector) retrieval (Zege et al., 2015). To compare with DNN_MPF and UH_MPFv2, we calculated the 8-day averages of the UB_MPF corresponding to the date ranges of the MODIS 8-day composite. All the MPFs are the fraction relative to the 12.5 km polar stereographic grid.

Figures 9-11 show the averaged DNN_MPF, UH_MPFv2 and UB_MPF in the period of May to September from 2003-2011 (note: the overlapping period of the three datasets is 2003-2011). Here we only consider the grids with SIC above 30% (Note: the DNN_MPF is restricted using NASA Team SIC and the UH_MPFv2, UB_MPF are restricted using the SIC by 100 minus the open water fraction in UH_MPFv2 dataset). In general, the climatology of the three MPFs are within 40%. In May, the DNN_MPF and UH_MPFv2 have similar pattern in much of the Arctic Ocean, although the DNN_MPF is relatively larger in the sea ice edge zone. The UB_MPF is generally larger than the other two data in the central Arctic. In June, the three data have comparable MPF in the ice edges (around 20-25%), especially in the Baffin Bay, Greenland Sea and Kara Sea. In the Arctic Basin, the DNN_MPF tends to evolve early in the eastern Arctic dominated by the first-year ice, while the UH_MPFv2 and UB_MPF seem to evolve early in the western Arctic. The MPF of all three data increases quickly in the bands of the Beaufort, Chukchi and East Siberian Seas. In July, higher fractions (above 25%) gradually extend to the central Arctic for the three data. The DNN_MPF and UB_MPF have higher fractions in the eastern and western Arctic basin, respectively, while the UH_MPFv2 has similar amount of MPF in the two regions. In August, the MPF of the three data gradually decreases. The UH_MPFv2 is generally higher than the other two data. The UH_MPFv2 and UB_MPF have the slowest and fastest decrease rate, respectively, while the decrease rate of the DNN_MPF is in between. The DNN_MPF and UH_MPFv2 have longer durations of high fractions than UB_MPF in the ice edges, especially for UH_MPFv2 with fraction above 25% for most areas until late August. By the end of August, the DNN_MPF and UB_MPF are less than 20% in the Arctic basin, while the UH_MPFv2 still maintains high fraction.

[Figure]

**Figure 9.** Averaged DNN_MPF relative to grid for the period of May to September from 2003-2011

[Figure]

**Figure 10.** Same as figure 9, except for the UH_MPFv2

[Figure]

**Figure 11.** Same as figure 9, except for the UB_MPF

Figure 12 shows the time series of the three MPFs relative to sea ice (note: we only consider the grids with SIC above 30% during 2003-2011). The DNN_MPF was transformed from the fraction relative to grid to sea ice using NASA Team SIC; the UH_MPFv2 and UB_MPF were transformed from the fraction relative to grid to sea ice using the UH_MPFv2 SIC (100 minus open water fraction in UH_MPFv2). The three datasets have similar pond fraction in early May (8.5%, 9.7 and 10.7% for DNN_MPF, UH_MPFv2 and UB_MPF). The MPFs of DNN_MPF and UB_MPF grow relatively quicker and are relatively larger than that of UH_MPFv2 until the end of June. From early July, the UH_MPFv2 always have higher MPF than that of the other two data. The DNN_MPF, UH_MPFv2 and UB_MPF reach the largest MPF ~27%, ~28% and ~27% in late July, the end of July, and early July, respectively. The MPF of UB_MPF decreases about two weeks earlier than that of the other two data. In August, the MPF of DNN_MPF and UB_MPF decrease relative faster than that of the UH_MPFv2. The UH_MPFv2 maintains high values (above 25%) for a longer duration than that of the other two data. The standard deviations of the three data are larger in August.

[Figure]

**Figure 12.** The evolution of the averaged three MPFs relative to sea ice in the period of May to September during 2003 to 2011. The error bars are the standard deviations during 2003-2011.

**6)** We provided the scatter plots with retrieved and observed MPF. To check the difference of the retrieved MPF from the network trained by 8-day composite of MODIS (hereafter referred to as DNN_8dayMODIS) and daily MODIS (hereafter referred to as DNN_dailyMODIS) surface reflectance. We further trained the network using the daily MODIS surface reflectance from MOD09GA. We compared the results with the MPF retrieved by DNN_8dayMODIS on the 500 m polar stereographic grid (Fig. 13 and 14). Note: the retrieved MPF is from DNN_MPF+NASASIC. (see details in **section 12)**). The results show that the retrieved MPF from both DNN_8dayMODIS and DNN_dailyMODIS have good relationship against the observations from HOTRAX (r = 0.63 and r = 0.69 for DNN_8dayMODIS and DNN_dailyMODIS) and NSIDC (r = 0.77 and r = 0.82 for DNN_8dayMODIS and DNN_dailyMODIS). The correlation with PRIC-Lei is better in DNN_8dayMODIS (r = 0.64 and r = 0.50 in DNN_8dayMODIS and DNN_dailyMODIS). The retrieved MPF from DNN_8dayMODIS and DNN_dailyMODIS have weak relationship with the observations from DLUT, TransArc and NPI. Overall, the performances of the retrieved MPF against the observations are generally consistent between DNN_8dayMODIS and DNN_dailyMODIS. This suggests that the networks trained by 8-day composite of MODIS surface reflectance is reliable in our study. The RMSE in DNN_8dayMODIS and DNN_dailyMODIS are generally within 0.1, which was proposed as an important factor to evaluate the data accuracy in Wright and Polashenski (2020).

[Figure]

**Figure 13.** Validation of the retrieved MPF against the observed MPF used in network training on the 500 m polar stereographic grid. (a) HOTRAX, (b) DLUT, (c) TransArc, (d) PRIC-Lei, (e) NSIDC, (f) NPI.

[Figure]

**Figure 14.** Same as Figure 13, except for the retrieved MPF from the network trained by the daily MODIS surface reflectance of MOD09GA.

Figure 15 shows the scatter plot of the MPF from DNN_MPF+NASASIC and the MPF version2 from University of Hamburg (https://icdc.cen.uni-hamburg.de/1/daten/cryosphere/arctic-meltponds.html, hereafter referred to as UH_MPFv2) against the observations on 12.5 km polar stereographic grid. We only use the observations where DNN_MPF and UH_MPFv2 are both within valid ranges. The DNN_MPF and DNN_MPFdailyMODIS on 12.5 km polar stereographic grid are both retrieved from the 8-day composite of MODIS (MOD09A1). Note: the DNN_MPF and DNN_MPFdailyMODIS are retrieved using the above-mentioned networks of DNN_8dayMODIS and DNN_dailyMODIS, respectively. The MPF from UH_MPFv2 is missing in validation with NSIDC and NPI (note: the correlation coefficients 0.53

and RMSE 0.107 of UH_MPF with NSIDC in Fig.4 TCD manuscript used the values in Rösel et al. (2012). The UH_MPF in Rösel et al. (2012) is version 1). The results show that the DNN_MPF has better agreement with the observations than DNN_MPFdailyMODIS. The completely independent validation with the observations from Webster (r = 0.63 and r = 0.51 for DNN_MPF and DNN_MPFdailyMODIS) and JOIS (r = 0.50 and r = 0.44 for DNN_ MPF and DNN_ MPFdailyMODIS) shows that the network trained using 8-day composite of MODIS is more robust. This further suggests that our MPF retrieval is reliable.

[Figure]

**Figure 15.** Validation of the MPFs against the observed MPF on the 12.5 km polar stereographic grid. (a) HOTRAX, (b) DLUT, (c) TransArc, (d) PRIC-Lei, (e) NSIDC, (f) NPI, (g) Webster, (h) JOIS.

**7)** We provided case studies of the observed MPF overlaid on the daily MODIS (MOD09GA) image with the original resolution (Fig.16-18). The MODIS images are generated using bands 1 (red), 4 (green) and 3 (blue). Note: we already provided the observed MPF overlaid on the NASA Team SIC in Fig.1-7.

We provided case studies of the observed MPF (original resolution) overlaid on the retrieved MPF of 12.5 km polar stereographic grid (Fig.19-24). The results show that the retrieved MPF generally agrees with the average of the observations in the same grid.

[Figure]

**Figure 16.** Observed MPF from DLUT on 20 August 2008 overlaid on the MODIS image.

[Figure]

**Figure 17.** Observed MPF from PRIC overlaid on the MODIS image. (a) observed MPF during (a) 28 July 2010, (b) 5 August 2010 and (c) 12 August 2012.

[Figure]

**Figure 18.** Observed MPF from NPI on 3 August 2012 overlaid on the MODIS image.

[Figure]

**Figure 19.** Observed MPF from HOTRAX overlaid on retrieved MPF from DNN_MPF+NASASIC.

[Figure]

**Figure 20.** Observed MPF from DLUT overlaid on retrieved MPF from DNN_MPF+NASASIC.

[Figure]

**Figure 21.** Observed MPF from PRIC-Lei overlaid on retrieved MPF from DNN_MPF+NASASIC.

[Figure]

**Figure 22.** Observed MPF from NSIDC overlaid on retrieved MPF from DNN_MPF+NASASIC.

[Figure]

**Figure 23.** Observed MPF from NPI overlaid on retrieved MPF from DNN_MPF+NASASIC.

[Figure]

**Figure 24.** Observed MPF from JOIS overlaid on retrieved MPF from DNN_MPF+NASASIC.

**8)** We provided the melt progress of the retrieved MPFs (DNN_MPF, UH_MPFv2 and UB_MPF) and the observed MPFs (Fig. 25). Note: the MPFs here are the fraction relative to grid. We use the average of the observations from HOTRAX, DLUT, TransArc, PRIC-Lei, NSIDC and NPI to represent for the observed MPF on the specific date range. We generally divide the melt progress every five days or more and estimate the average of the observed MPF and the corresponding retrieved MPFs within the days. Note: if the corresponding retrieved MPFs was missing, we use the average of the retrieved MPFs during 2003-2011 within the days. The results show that the DNN_MPF and the UH_MPFv2 are closer to the observed MPFs in May during the early melting season. Then the DNN_MPF shows better agreements with the observed MPFs during early to mid-June. The three retrieved MPFs in July are close and show good agreements with the observed MPFs. In later melting season, the UH_MPFv2 and UB_MPF are respectively larger and smaller than the observed MPFs. The DNN_MPF in later melting season is generally within the range of the UH_MPFv2 and UB_MPF and is closer to the observed MPFs.

[Figure]

**Figure 25.** Evolution of the MPF relative to grid from the retrieved and observed MPF. The x-axis is the melt progress divided by around every five days or more in the period of May to September (i.e., 6.01-6.05 is the period of 1-5 June).

**9)** We provided the detailed data processing in **section 4)**. For each 8-day composite of MOD09A1, we have 40 tiles (h09v02-h26v02, see https://modis-land.gsfc.nasa.gov/MODLAND_grid.html for details) in total to cover the entire Arctic. We mosaiced all the tiles into one *.hdf file using the MODIS Reprojection Tool (MRT) and then reprojected the mosaic to a GeoTIFF on the 500 m polar stereographic grid using ArcGIS. Each band (band 1, 2, 3 and 5) of the MOD09A1 was stored as a separate GeoTIFF file. For the network training, the input is the surface reflectance from the GeoTIFF files of the four bands on the 500 m polar stereographic grid.

For the observed MPFs from each source, we use the corresponding latitude and longitude to determine which gird cell (500 m polar stereographic grid) the observation falls in. If more than one observation from one source on a specific day fall in the same 500 m polar stereographic grid, the average of those observations is used as the training (target) data in the network. Note: the observed MPF relative to sea ice area has been transformed to the MPF relative to the grid (image area or coverage of each observation) based on the observed SIC in the network training.

**10)** We provided the accuracy of our dataset using the uncertainties of the MPF (DNN_MPF) and SIC (DNN_SIC) retrieved from DNN_MPF+NASASIC (see **section 12)** for details). Note: in this revision, we only consider the girds with SIC above 30%. The uncertainties are estimated by the standard deviations among the outputs of networks within 10-90 percentile of the 100 networks (described in line 172-173 in TCD manuscript). Table 1 shows the average standard deviations of the DNN_MPF and DNN_SIC for the period of May to August during 2000-2017. The results show that the magnitude of the uncertainty of our MPF retrieval varies slightly from May to August, with an average of 3.1%. The uncertainty of SIC is relatively larger than that of the MPF, within 4-5% before August and then increases by 1-2% in August and

September. Figure 26 shows the spatial distribution of the standard deviations of DNN_MPF averaged for the period of May to September during 2000-2017. The uncertainties of MPF are generally within 4% in much of the Arctic, except for the Canadian Arctic in mid-June (~7%). The MPF in the ice edges does not show large uncertainties. This suggests our MPF retrieval is reliable.

**Table 1.** The uncertainties of the MPF and SIC from the DNN_MPF+NASASIC

| Date | MPF uncertainty (%) | SIC uncertainty (%) |
|---|---|---|
| 05/09 | 3.04 | 4.04 |
| 05/17 | 3.08 | 4.11 |
| 05/25 | 3.26 | 4.31 |
| 06/02 | 3.37 | 4.27 |
| 06/10 | 3.66 | 4.61 |
| 06/18 | 3.37 | 4.60 |
| 06/26 | 2.95 | 4.52 |
| 07/04 | 2.56 | 4.14 |
| 07/12 | 2.39 | 4.12 |
| 07/20 | 2.39 | 4.26 |
| 07/28 | 2.61 | 4.68 |
| 08/05 | 2.86 | 5.14 |
| 08/13 | 3.16 | 5.64 |
| 08/21 | 3.33 | 5.77 |
| 08/29 | 3.72 | 6.18 |
| 09/06 | 4.03 | 6.41 |
| Average | 3.11 | 4.80 |

[Figure]

**Figure 26.** The average uncertainties of the MPF retrieved from DNN_MPF+NASASIC in the period of 2000-2017

**11)** We further provided the spatial distribution of standard deviation of the DNN_MPF and DNN_SIC in 2004 and 2012 (Fig. 27-30). Note: in the revision, we only consider the gird with SIC above 30%, instead of 15%. The results show that the uncertainties of DNN_MPF in 2004 and 2012 are within 4% in most areas, except for the DNN_MPF in the Canadian central Arctic in June. The uncertainties of DNN_MPF is a little bit larger in 2004 than in 2012. It should be noted that the DNN_MPF does not show large uncertainties in the ice edges during most periods, except for the DNN_MPF in early August in 2012, which has uncertainties around 6-8% in the ice edges of Beaufort and Chukchi Seas. The uncertainties of the DNN_SIC is larger than that of the DNN_MPF in 2004 and 2012. The largest uncertainties of DNN_SIC appear in August.

[Figure]

**Figure 27.** The uncertainties of the DNN_MPF in 2004

[Figure]

**Figure 28.** The uncertainties of the DNN_SIC in 2004

[Figure]

**Figure 29.** The uncertainties of the DNN_MPF in 2012

[Figure]

**Figure 30.** The uncertainties of the DNN_SIC in 2012

**12)** To further address the concern, here we added observed SIC as the target data in the network training, and also retrieved SIC as the second output. We used the observed SIC from three independent sources as the target and trained the network separately. (note: the first output is MPF, the same as described in section 2 of TCD manuscript). Table 2 provides the detailed information.

**Table 2.** Details of the target and output for the network

| Network | Training Input | Training | Output (target) |
|---|---|---|---|
| DNN_MPF (no SIC) | | Observed MPF | MPF (no SIC) |
| DNN_MPF+NASASIC | MOD09A1 bands (Band 1, 2, 3, 5) | Observed MPF & NASA Team SIC | |
| DNN_MPF+FieldSIC | | Observed MPF & Observed SIC | MPF + SIC |
| DNN_MPF+AMSRSIC | | Observed MPF & AMSR-SIC | |

- DNN_MPF (no SIC) is the network trained in the TCD manuscript. The training input is the four MOD09A1 bands (Band 1, 2, 3, 5) on the 500 m polar stereographic grid. The training output is the observed MPF from six sources (HOTRAX, DLUT, TransArc, PRIC-Lei, NSIDC, NPI, see detailed information about the observed MPF in **section 2)**). The DNN_MPF (no SIC) does not include SIC as the target in the network training.

- DNN_MPF+NASASIC is the network trained by adding the NASA Team SIC

(Cavalieri et al., 1996) as the second target. The NASA Team SIC is derived from Nimbus-7 SMMR and DMSP SSM/I-SSMIS Passive Microwave Data using a revised NASA Team algorithm (https://nsidc.org/data/nsidc-0051). In the network training, the NASA Team SIC was resampled from 25 km to the 500 m polar stereographic grid to match the resolution of the MODIS surface reflectance.

- DNN_MPF+FieldSIC is the network trained by adding the observed SIC from multi-sources (HOTRAX, DLUT, TransArc, PRIC-Lei, NSIDC and NPI) as the second target. The observed SIC is obtained from the same sources as the observed MPF. In the network training, the observed SIC was resampled from its original resolution (coverage) to the 500 m polar stereographic grid to match the resolution of MODIS surface reflectance (note: we use the average of the observed SIC from each source located in the same grid as the resampled SIC).

- DNN_MPF+AMSRSIC is the network trained by adding the SIC derived from Advanced Microwave Scanning Radiometer-Earth Observing System and Advanced Microwave Scanning Radiometer 2 (hereafter referred to as AMSR SIC, Spreen et al., 2008) as the second target. The AMSR SIC is developed by the University of Bremen using the ARTIST Sea Ice (ASI) algorithm (https://seaice.uni-bremen.de/sea-ice-concentration). In the network training, the AMSR SIC was resampled from 6.25 km to the 500 m polar stereographic grid to match the resolution of MODIS surface reflectance.

For the final MPF and SIC data retrieval, the data on the 12.5 km polar stereographic grid were used in the ensemble-based network (note: MOD09A1 on the 12.5 km polar stereographic grid was used as the input). The only difference between DNN_MPF (no SIC) and the other three networks (DNN_MPF+NASASIC, DNN_MPF+FieldSIC and DNN_MPF+AMSRSIC) is that the three networks contain SIC as the second target in network training. Therefore, the final dataset from DNN_MPF (no SIC) only contains MPF on the 12.5 km polar stereographic grid and the final dataset from the other three networks contains MPF and SIC on the 12.5 km polar stereographic grid.

Figure 31 shows the correlation coefficients and the RMSE of MPF from the above four network training. It appears that the correlation coefficients of the four networks with independent SIC are comparable. This is also true for the RMSE. This suggests that the influence of the ice concentration on the retrieved MPF is minor. This further increases the reliability of our MPF retrieval. We check the spatial correlation coefficients and RMSE of the MPF from three re-trained networks with the MPF from DNN_MPF (no SIC) in each year during 2000-2017. The results show that the average spatial correlation coefficient is ~0.99 and the RMSE is ~0.012. This suggests that the MPF from the re-trained networks are generally consistent with that from DNN_MPF (no SIC).

[Figure]

**Figure 31.** Validation of the MPF from four networks against the observed MPF: (a) correlation coefficients and (b) RMSE. (repetition of Fig.4 in the TCD manuscript).

For further comparison, we show the MPF (relative to grid) in 2017 from DNN_MPF (no SIC) and the three re-trained networks (DNN_MPF+NASASIC, DNN_MPF+FieldSIC and DNN_MPF+AMSRSIC). The results show that the spatial MPF during May to September in 2017 from DNN_MPF (no SIC) (Fig.32) are almost the same with that from the three networks added SIC (Fig.33 to 35). This further suggests that the SIC only has very limited effect on the MPF retrieval in our method.

[Figure]

**Figure 32.** The evolution of the MPF from DNN_MPF (no SIC) relative to grid from early May to early September in 2017.

[Figure]

**Figure 33.** Same as Fig.32, except for the MPF from DNN_MPF+NASASIC.

[Figure]

**Figure 34.** Same as Fig.32, except for the MPF from DNN_MPF+FieldSIC.

[Figure]

**Figure 35.** Same as Fig.32, except for the MPF from DNN_MPF+AMSRSIC.

Table 3 shows the percentage of grid cell with MPF greater than SIC (regarded as bad retrieval). The MPF (relative to grid) and SIC used here are both from the three re-trained networks (DNN_MPF+NASASIC, DNN_MPF+FieldSIC and DNN_MPF+AMSRSIC). The results show that 0.84-1.31% of the grid cells have bad MPF retrieval when considering grid cell with SIC>15%. It can be reduced to 0.05-0.19% of the grid cells when considering SIC>30%. The bad retrieval (MPF larger than SIC) has been removed in the analyses. Compared to Table 1 in the preliminary response to the review#1, the percentage of the grid with MPF larger than SIC does not change much whether the MPF is from DNN_MPF (no SIC) or the three re-trained networks (note: 1.97% and 0.09% of the grid cells have bad MPF retrieval when considering grid cell with SIC>15% and SIC>30% in DNN_MPF (no SIC)). This suggests that the SIC has very limited effect on the MPF retrieval in our method, which further increases the reliability of our method.

In order to minimize the bad MPF retrievals that are primarily located in the sea ice edge area with small concentration. In this revision, we only consider the grid cell with sea ice concentration greater than 30%, instead of 15%. The original MPF from DNN_MPF (no SIC) has been replaced by the retrieval from DNN_MPF+NASASIC.

**Table 3**. The percentage of the grid cell with MPF relative to grid greater than SIC

| Year | MPF > Retrieved SIC | | | | | | Total grids |
| | DNN_MPF+NASASIC | | DNN_MPF+FieldSIC | | DNN_MPF+AMSRSIC | | |
| | SIC>15% | SIC>30% | SIC>15% | SIC>30% | SIC>15% | SIC>30% | |
|---|---|---|---|---|---|---|---|
| 2000 | 1.85 | 0.17 | 1.29 | 0.08 | 1.14 | 0.27 | 49127 |
| 2001 | 1.45 | 0.13 | 0.92 | 0.03 | 0.77 | 0.22 | 45253 |
| 2002 | 1.30 | 0.10 | 1.02 | 0.04 | 0.84 | 0.21 | 47358 |
| 2003 | 1.50 | 0.13 | 1.19 | 0.07 | 1.01 | 0.18 | 48097 |
| 2004 | 1.29 | 0.12 | 0.96 | 0.04 | 0.96 | 0.20 | 47545 |
| 2005 | 1.46 | 0.12 | 1.18 | 0.06 | 1.00 | 0.22 | 45805 |
| 2006 | 1.60 | 0.12 | 1.21 | 0.06 | 1.05 | 0.25 | 45281 |
| 2007 | 1.49 | 0.11 | 1.21 | 0.05 | 1.04 | 0.20 | 42082 |
| 2008 | 1.52 | 0.11 | 1.41 | 0.09 | 1.18 | 0.16 | 43445 |
| 2009 | 1.71 | 0.13 | 1.44 | 0.09 | 1.24 | 0.22 | 44937 |
| 2010 | 1.50 | 0.12 | 1.17 | 0.04 | 0.86 | 0.22 | 42775 |
| 2011 | 1.42 | 0.10 | 1.21 | 0.05 | 0.86 | 0.19 | 41503 |
| 2012 | 0.93 | 0.09 | 0.91 | 0.03 | 0.57 | 0.11 | 39476 |
| 2013 | 1.05 | 0.08 | 0.85 | 0.02 | 0.44 | 0.13 | 43269 |
| 2014 | 1.23 | 0.09 | 1.05 | 0.05 | 0.99 | 0.17 | 43127 |
| 2015 | 0.66 | 0.07 | 0.57 | 0.01 | 0.33 | 0.11 | 41843 |
| 2016 | 0.87 | 0.08 | 0.62 | 0.02 | 0.42 | 0.16 | 40403 |
| 2017 | 0.82 | 0.08 | 0.78 | 0.05 | 0.49 | 0.11 | 41081 |
| Average | 1.31 | 0.11 | 1.06 | 0.05 | 0.84 | 0.19 | 44023 |

*"You motivate (in Sect. 2.1) the inclusion of a fourth spectral band with the fact that by this action you are able to better discriminate property changes within the snow pack. While this might be an advantage for the early phase of melt (which you could have explained in more detail) it seems not to be clearly stated how this could improve melt-pond fraction retrieval at a later stage. I guess, one of the main suggestions for improvement in the Rösel et al. paper was motivated by the change in spectral characteristics over the course of the melt season resulting in a different spectral response of melt ponds on MYI compared to melt ponds on FYI. This is where I hoped that your paper would enhance the current state-of-the-art but I have difficulties to see this in the paper yet."*

**Response:**

**13)** In the revision, we provided the explanation why we added band 5 in this study. According to Barber et al., 1992, "Spectral albedos collected over snow surfaces during SIMS '90 indicate minimal variation in reflectance throughout the visible portion of the electromagnetic spectrum. Reflectance decreased within the near-infrared, illustrating the wavelength dependence and sensitivity of snow reflectance to phase changes within the snow cover. A temperature increase of 5.5C° promoted a phase change within the

snow pack from ice to liquid and vapour, which caused the associated changes in grain size (increase) and structure (rounding). Spectral albedo in the near-infrared region is most sensitive to these changes". Thus, we added one more near-infrared band (band5) in the MPF retrieval that may detect the changes in the stage of snow melting.

In this revision, we estimated the contribution of the four MODIS bands to the MPF and SIC retrieval in the network training (Fig. 36). The contribution was estimated based on "Connection weights" following Olden and Jackson (2002). The "Connection weights" calculates the product of the raw input-hidden and hidden-output connection weights between each input neuron and output neuron and sums the products across all hidden neurons. The results show that the band 5 accounts for ~20% of the retrieval of MPF and SIC, although its contribution is relatively less than the other three bands. This further suggests that adding band 5 benefits the retrieval.

[Figure]

**Figure 36.** Contribution of the MODIS bands in network training.

**14)** We provided the explanation of the two improvements in this study. i) We added one near-infrared band (band 5) in the study to extend the bandwidth over 1000 nm. Figure 36 shows the band 5 accounts for about 20% contribution to the retrieval of MPF, which means adding this band benefits the retrieval. The previous research also showed that spectral albedo in the near-infrared region is more sensitive in the changes within the snow pack from ice to liquid and vapour. ii) Compared to the UH_MPFv2, we did not use the fixed spectral reflectance of each surface type (i.e., bare ice, snow covered ice, melt pond and open water) to build the relationship. Instead, we used the observed MPF from multi-sources to directly train the deep neural network. The advantage of our method is that it avoids large uncertainties of spatially and temporally varying reflectance associated with different surface species, which can result in large uncertainties for the retrieval of type fraction.

More recently, Wright and Polashenski (2020) compared the MPF retrieval using the "spectral unmixing" method in Rösel et al. (2015) and a random forest machine learning model which does not rely on the constant spectral reflectance of each surface type. Their results suggested that "it is not possible to consistently derive component surface fractions of sea ice from low resolution imagery using spectral unmixing techniques at an accuracy suitable for validating melt pond models or establishing unambiguous long

term trends." Moreover, the "spectral unmixing" is "highly sensitive to error in the input surface reflectance data". Their tests show "the accuracy of the machine learning method to be better than regionally tuning spectral unmixing and it is significantly more feasible to implement.". The RMS error in melt pond determination could be improved from 0.18 they found in spectral unmixing techniques to 0.07 using machine learning. This further suggests that our method based on network training to retrieve the MPF is feasible.

"*Lines 120-122: You use a standard sea-ice concentration product as a sea-ice mask. While this is fine, several questions immediately pop up: i) what is meant by "revised NASA Team algorithm (Cavalieri et al., 1996)"? The year of the reference makes clear that it cannot be the enhanced NASA Team algorithm". ii) what are the specifications of this data set in terms of spatial and temporal resolution and how did you pre-process the data to match with the MODIS data? iii) passive microwave concentration have biases during summer as has been discussed, e.g., in Comiso and Kwok in 1996: "Surface and radiative characteristics of the summer Arctic sea ice cover from multisensory satellite observations" and in Kern et al. in 2016: "The impact of melt ponds on summertime microwave brightness temperatures and sea-ice concentrations". Doesn't using such sea-ice concentration data sets as sea-ice mask therefore require a more in-depth description of how you used the data and how the expected bias in sea-ice concentration influences your melt-pond retrieval?*"

**Response:**

**15)** We provided the explanation about the "revised NASA Team algorithm". In the section of "User Guide" at https://nsidc.org/data/nsidc-0051, it says "Sea ice concentrations for this data set were produced using a revised NASA Team algorithm that uses a different set of tie points and weather filters than the original NASA Team algorithm. The NASA Technical Memorandum 104647 (Cavalieri et al., 1997) includes information about differences, such as tie points, between the original algorithm and the revised NASA Team algorithm." In the section of "Citing These Data", it says that we should cite the data as "Cavalieri, D. J., C. L. Parkinson, P. Gloersen, and H. J. Zwally. 1996, updated yearly. Sea Ice Concentrations from Nimbus-7 SMMR and DMSP SSM/I-SSMIS Passive Microwave Data, Version 1. [Indicate subset used]. Boulder, Colorado USA. NASA National Snow and Ice Data Center Distributed Active Archive Center. doi: https://doi.org/10.5067/8GQ8LZQVL0VL." Since our final MPF dataset retrieved from DNN is the fraction relative to grid on 12.5 km polar stereographic grid, the NASA Team SIC is also resampled from 25 km (original resolution) to 12.5 km polar stereographic grid to match the grid size of the MPF.

**16)** According to **section 12)**, the MPFs retrieved from the networks that included or excluded the SIC as the target data in training are very close. This suggests that sea ice concentration has minor effect on the retrieval of MPF in this study. To further check the uncertainty due to the low ice concentration in the ice edges, we provided the

comparison of the MPF relative to sea ice by using a) the NASA Team SIC and b) adjusted NASA Team SIC based on Kern. et al. (2016). According to the Table 12 in Kern. et al. (2016), the SIC retrieved from NASA Team algorithm in CaseA60 and CaseA80 (Cases A60 and A80 denote 100% sea-ice concentration with 40 and 20% (apparent) open-water fraction due to melt ponds) are underestimated by 22.6 and 0% SIC, respectively. We add 22% and 10% SIC to the NASA Team SIC (hereafter referred to as adjusted SIC), where the MPF is above 40% and 30%. We only consider the grids with SIC above 30% during 2000-2017. Figure 37 shows the evolution of the MPF relative to sea ice and adjusted sea ice (note: the MPF is from the DNN_MPF+NASASIC). The results show that the underestimation of SIC in the ice edges due to the presence of melt ponds has minor effect on the evolution of MPF to ice-covered area. This could be explained by the limited percentage ~0.1% and ~2.65% of the grids (SIC above 30%) with MPF (relative to grid) above 40% and 30% shown in Table 4. This suggests that the potentially affected grids only have small amounts in our study, and those grids will not change the major results.

[Figure]

**Figure 37.** The evolution of the averaged MPF to sea ice and adjusted sea ice from May to September during 2000 to 2017.

**Table 4.** Ratios of the grids with MPF relative to grid above 40% and 30%

| Year | Ratio (%) of the grids with MPF above 40% | Ratio (%) of the grids with MPF above 30% | Total grids |
|---|---|---|---|
| 2000 | 0.07 | 2.81 | 49127 |
| 2001 | 0.17 | 3.04 | 45253 |
| 2002 | 0.15 | 2.54 | 47358 |
| 2003 | 0.13 | 3.44 | 48097 |
| 2004 | 0.10 | 2.00 | 47545 |
| 2005 | 0.09 | 2.59 | 45805 |
| 2006 | 0.14 | 2.48 | 45281 |
| 2007 | 0.09 | 4.58 | 42082 |
| 2008 | 0.09 | 2.82 | 43445 |
| 2009 | 0.12 | 2.45 | 44937 |
| 2010 | 0.08 | 2.55 | 42775 |
| 2011 | 0.07 | 3.59 | 41503 |
| 2012 | 0.09 | 2.81 | 39476 |
| 2013 | 0.08 | 1.74 | 43269 |
| 2014 | 0.09 | 2.87 | 43127 |
| 2015 | 0.05 | 1.74 | 41843 |
| 2016 | 0.05 | 2.06 | 40403 |
| 2017 | 0.14 | 1.52 | 41081 |
| Average | 0.10 | 2.65 | 44023 |

*"It appears to me that you did not yet adequately cite the MODIS melt-pond fraction data set of Rösel et al. (2012) which you are using in your overall comparison (e.g. Figure 4). Would you mind to check which version of this data set you used and provide the doi and version of it in your reference list? I guess this would help other potential users to locate the correct data set."*

*"Lines 148-153: I checked the Webster et al. [2015] paper. I have serious doubts that this is the correct reference. I found that this paper basically compares a new method to derive melt-pond fraction based on APLIS campaign data and compared the results with SHEBA data. I did not find the mentioned 2000-2014 MPF data set. Here you would appreciate a hint about where to find this potentially very valuable data set."*

**Response:**

**17)** The MODIS melt pond fraction dataset of Rösel et al. (2015) used in our study is version 2 (referred to as UH_MPFv2 in the response), which is obtained from the Integrated Climate Data Center (ICDC) (https://icdc.cen.uni-hamburg.de/1/daten/cryosphere/arctic-meltponds.html). The doi is "Rösel, Anja; Kaleschke, Lars; Kern, Stefan (2015). Gridded Melt Pond Cover Fraction on Arctic Sea Ice derived from TERRA-MODIS 8-day composite Reflectance Data bias corrected Version 02. World Data Center for Climate (WDCC) at DKRZ. https://doi.org/10.1594/WDCC/MODIS__Arctic__MPF_V02". The method of the MPF retrieval is described in "Rösel, A., Kaleschke, L., & Birnbaum, G. (2012). Melt ponds on Arctic sea ice determined from MODIS satellite data using an artificial neural

network. The Cryosphere, 6, 431-446." In the revision, we added these information.

**18)** The Webster's MPF dataset can be found at "http://psc.apl.uw.edu/melt-pond-data/". According to the data description, "the data set is generated from previously classified high resolution visible band satellite images following Webster et al. 2015. The data set contains two separate sets, one covering the periods of 1999-2014 which was derived by Melinda (see Webster et al. 2015 for details), the other was derived by Florence Fetterer (NSIDC) using a supervised classification technique and covers the period 1999-2001". In the validation, we only use the first dataset. Based on the "How to cite" in the page, the data is cited as "Webster, M. A., I. G. Rigor, D. K. Perovich, J. A. Richter-Menge, C. M. Polashenski, and B. Light (2015), Seasonal evolution of melt ponds on Arctic sea ice, J. Geophys. Res. Oceans, 120, doi:10.1002/2015JC011030." in our manuscript. We provided the data link in the revision.

Here we provided more description of the observation from Webster. The MPFs were retrieved by the Polar Science Center, University of Washington based on the classified high-resolution visible band satellite images following Webster et al. (2015). The image data source has been referred to as Global Fiducial Imagery, Literal Image Derived Products, National Technical Means images, and MEDEA Measurements of Earth Data for Environmental Analysis. The MPFs were measured at 69-86.5°N over the Beaufort Sea, Chukchi Sea, the Canadian Arctic, the Fram Strait, and the East Siberian Sea from May to August for the period of 1999-2014. The scene size (square grid) of the MPFs ranges from 5 to 25 km. The obtained measurement from Webster is the MPF relative to sea ice cover. In validation, the MPF has been transferred to the fraction relative to the grid (image area) using the measured SIC from Webster. The data and detailed description can be found at http://psc.apl.uw.edu/melt-pond-data/.

"*Figure 5: This figure states an average (2000-2017) pan-Arctic melt-pond fraction of 10% already in the middle of May. This appears to be too large. While Liu et al. (2015) found a similar evolution they used the old Rösel et al. melt-pond fraction data set which was erroneously high and which has been corrected based on the findings presented in Mäkynen et al. (2014). As you state yourself in the paper, melt onset typically occurs early June and I'd even state the melt onset for the majority of the MYI is in late June / early July which is when you suggest a melt-pond fraction over MYI of 15% already.*"

**Response:**

**19)** In this revision, we only consider the grids with SIC above 30% instead of 15%. The MPF relative to sea ice during 2000-2017 decreases from 8.4% to 7.8% in early May by considering SIC above 30%. According to Fig.12, the averaged MPFs relative to sea ice during 2003-2011 from the three datasets (DNN_MPF, UH_MPFv2, UB_MPF) are close in May. The DNN_MPF is about 1.2% and 0.4% smaller than the UH_MPFv2 in early and mid-May. Therefore, we think the DNN_MPF in the middle of May is acceptable.

Here we plotted the observed MPF (relative to sea ice) from NSIDC on FYI and MYI

(Fig. 38). The sea ice age data is obtained from the EASE-Grid Sea Ice Age, Version 4 (https://nsidc.org/data/nsidc-0611). Figure 38 shows that some of the observed MPF on 25 June is already above 25% on FYI and MYI, and the averages are 19% and 16%, respectively. In our study, the DNN_MPF (retrieved from DNN_MPF+NASASIC) relative to sea ice on MYI is around 4% and 10% in early May and early June, and increases to around 16% in late June, which matches with the observations from NSIDC in the same period. Therefore, according to the observed MPF from NSIDC on MYI, we think it is possible that the MPF relative to sea ice on MYI can reach to 15% in late June.

[Figure]

**Figure 38.** The observed MPF relative to sea ice from NSIDC on FYI and MYI.

"*Figures 10 and 11: These are 8-daily estimates of the melt-pond fraction. How come that compared to the Rösel et al. product there are no gaps due to clouds? It appears to be very unlikely that the more recent collection of MODIS data you used does contain less pixels flagged as cloud covered.*"

**Response:**

**20)** The UH_MPFv2 from Rösel et al. (2015) was retrieved from the MODIS Surface Reflectance 8-Day L3 Global 500m SIN Grid V005. The DNN_MPF in this study was retrieved from the MODIS Surface Reflectance 8-Day L3 Global 500m SIN Grid V006. Both datasets are retrieved from the 8-day composite of MODIS and are on the 12.5 km polar stereographic grid. For validation against the observed MPF, we only consider the corresponding grids in which the two MPF data have valid values. Note: for UH_MPFv2, we use the parameter named as "mpf" (melt pond fraction at top of sea ice)" in the comparison and validation.

Since the MOD09A1 v005 is not available online, we cannot compare the detailed difference of the cloud mask between the version 5 and 6. However, the files of

"Collection 6 updates to the MODIS Cloud Mask (MOD35)" and "Collection 6 Cloud Mask (MOD35) Status and Recent Analysis" suggest the MODIS collection 6 has a more accurate cloud mask, which tends to reduce the cloud uncertainty. The two documents can be found at https://modis.gsfc.nasa.gov/sci_team/meetings/201001/presentations/atmos/frey.pdf, and
https://modis.gsfc.nasa.gov/sci_team/meetings/201105/presentations/atmos/frey.pdf

Reference:

Barber, D. G., Flett, D. G., De Abreu, R. A., and LeDrew, E. F.: Spatial and Temporal Variation of Sea Ice Geophysical Properties and Microwave Remote Sensing Observations: The SIMS'90 Experiment, Arctic, 45, 233-251, 1992.

Cavalieri, D. J., Parkinson, C. L., Gloersen, P., and Zwally, H. J.: Sea Ice Concentrations from Nimbus-7 SMMR and DMSP SSM/I-SSMIS Passive Microwave Data, Version 1 (updated yearly), NASA National Snow and Ice Data Center Distributed Active Archive Center, https://doi.org/10.5067/8GQ8LZQVL0VL, 1996.

Divine, D. V., Granskog, M. A., Hudson, S. R., Pedersen, C. A., Karlsen, T. I., Divina, S. A., Renner, A. H. H., and Gerland, S.: Regional melt-pond fraction and albedo of thin Arctic first-year drift ice in late summer, Cryosphere, 9, 255–268, https://doi.org/10.5194/tc-9-255-2015, 2015.

Divine, D. V., Pedersen, C. A., Karlsen, T. I., Aas, H. F., Granskog, M. A., Hudson, S. R., and Gerland, S.: Photogrammetric retrieval and analysis of small scale sea ice topography during summer melt, Cold Reg. Sci. Technol., 129, 77-84, https://doi.org/10.1016/j.coldregions.2016.06.006, 2016.

Fetterer, F., and Untersteiner N.: Observations of Melt Ponds on Arctic Sea Ice, J. Geophys. Res., 103 (C11), 24,821-24,835, https://doi.org/10.1029/98JC02034, 1998.

Fetterer, F., Wilds, S., and Sloan, J.: Arctic Sea Ice Melt ponds Statistics and Maps, 1999-2001, Version 1, NSIDC: National Snow and Ice Data Center, accessed 10 June 2018, https://doi.org/10.7265/N5PK0D32, 2008

Huang, W., Lu, P., Lei, R., Xie, H., and Li, Z.: Melt ponds distribution and geometry in high Arctic sea ice derived from aerial investigations, Ann. Glaciol., 57, 105-118, https://doi.org/10.1017/aog.2016.30, 2016.

Istomina, L., Heygster, G., Huntemann, M., Schwarz, P., Birnbaum, G., Scharien, R., Polashenski, C., Perovich, D., Zege, E., Malinka, A., Prikhach, A., and Katsev, I.: Melt pond fraction and spectral sea ice albedo retrieval from MERIS data – Part 1: Validation against in situ, aerial, and ship cruise data, The Cryosphere, 9, 1551-

1566, doi:10.5194/tc-9-1551-2015, 2015.

Kern, S., Rösel, A., Pedersen, L. T., Ivanova, N., Saldo, R., and Tonboe, R. T.: The impact of melt ponds on summertime microwave brightness temperatures and sea-ice concentrations, The Cryosphere, 10, 2217–2239, https://doi.org/10.5194/tc-10-2217-2016, 2016.

Lei, R., Tian-Kunze, X., Li, B., Heil, P., Wang, J., Zeng, J., and Tian, Z.: Characterization of summer Arctic sea ice morphology in the 135°–175° W sector using multi-scale methods, Cold Reg. Sci. Technol., 133, 108-120, https://doi.org/10.1016/j.coldregions.2016.10.009, 2017.

Lu, P., Li, Z., Cheng, B., Lei, R., and Zhang, R.: Sea ice surface features in Arctic summer 2008: Aerial observations, Remote Sens. Environ., 114, 693-699, https://doi.org/10.1016/j.rse.2009.11.009, 2010.

Nicolaus, M., Katlein, C., Maslanik, J. A., and Hendricks, S.: Sea ice conditions during the POLARSTERN cruise ARKXXVI/3 (TransArc) in 2011, PANGAEA Dataset, PANGAEA, Alfred Wegener Institute, Helmholtz Center for Polar and Marine Research, Bremerhaven, https://doi.org/10.1594/PANGAEA.803312, 2012b

NSIDC: SHEBA Reconnaissance Imagery, Version 1.0. Boulder, Colorado USA: National Snow and Ice Data Center. Digital media, 2000.

Olden, J. D., and Jackson, D. A.: Illuminating the "black box": a randomization approach for understanding variable contributions in artificial neural networks, Ecological modelling, 154(1-2), 135-150., https://doi.org/10.1016/S0304-3800(02)00064-9, 2002.

Perovich, D. K., Grenfell, T. C., Light, B., Elder, B. C., Harbeck, J., Polashenski, C., Tucker, W. B., and Stelmach, C.:Transpolar observations of the morphological properties of Arctic sea ice, J. Geophys. Res.: Oceans, 114, C00A04, https://doi.org/10.1029/2008JC004892, 2009.

Rösel, A., Kaleschke, L., and Birnbaum, G.: Melt ponds on Arctic sea ice determined from MODIS satellite data using an artificial neural network, Cryosphere, 6, 431-446, https://doi.org/10.5194/tc-6-431-2012, 2012.

Rösel, A., Kaleschke, L., Kern, S.: Gridded Melt Pond Cover Fraction on Arctic Sea Ice derived from TERRA-MODIS 8-day composite Reflectance Data bias corrected Version 02. World Data Center for Climate (WDCC) at DKRZ. https://doi.org/10.1594/WDCC/MODIS__Arctic__MPF_V02, 2015.

Spreen, G., Kaleschke, L., and Heygster, G.: Sea ice remote sensing using AMSR-E 89 GHz channels, J. Geophys. Res., 113, C02S03, https://doi.org/10.1029/2005JC003384, 2008.

Tanaka, Y., Tateyama, K., Kameda, T., and Hutchings, J. K.: Estimation of melt ponds fraction over high-concentration Arctic sea ice using AMSR-E passive

microwave data, J. Geophys. Res.: Oceans, 121, 7056-7072, https://doi.org/10.1002/2016JC011876, 2016.

Vermote, E.: MOD09A1 MODIS/Terra Surface Reflectance 8-Day L3 Global 500m SIN Grid V006 [Data set]. NASA EOSDIS Land Processes DAAC, accessed 2019-07-04 from https://doi.org/10.5067/MODIS/MOD09A1.006, 2015.

Vermote, E., Wolfe, R.: MOD09GA MODIS/Terra Surface Reflectance Daily L2G Global 1kmand 500m SIN Grid V006 [Data set]. NASA EOSDIS Land Processes DAAC, accessed 2019-12-28 from https://doi.org/10.5067/MODIS/MOD09GA.006, 2015.

Webster, M. A., Rigor, I. G., Perovich, D. K., Richter-Menge, J. A., Polashenski, C. M., and Light, B.: Seasonal evolution of melt ponds on Arctic sea ice, J. Geophys. Res.: Oceans, 120, 5968-5982, https://doi.org/10.1002/2015JC011030, 2015.

Wright, N. C., and Polashenski, C. M.: How machine learning and high-resolution imagery can improve melt pond retrieval from MODIS over current spectral unmixing techniques, J. Geophys. Res.: Oceans, e2019JC015569, https://doi.org/10.1029/2019JC015569, 2020.

---

## Author Comment (AC4) · 8 Mar 2020

**Response to the reviews of TC-2019-208 "Investigation of spatiotemporal variability of melt pond fraction and its relationship with sea ice extent during 2000–2017 using a new data" by Yifan Ding, Xiao Cheng, Jiping Liu, Fengming Hui, and Zhenzhan Wang**

We greatly appreciate the thoughtful comments from the reviewer. According to the reviewer's comments, we revised the original manuscript. All issues raised have been considered thoroughly.

**Comments by reviewer #2**

"a) The weak points of the manuscript are the network training and the validation. It is far too early to include MPF trend and MPF map analysis before these are sorted out, as well as claim to outperform another retrieval. The provided description of the in-situ training data, validation data, and validation results are insufficient and do not allow to assess the performance of the retrieval.

Please provide:

- a detailed description of the training and the validation datasets you use - current description is confusing and hard to understand. For each dataset, the size of the sample, spatial resolution, spatial coverage, temporal coverage, and the method of spatial and temporal collocation should be clearly stated. For each 8-day MODIS composite that you compare, how many days offset to in situ data do you allow? when you do that, do you have any assumptions about the evolution of MPF? how to do you train a neural network for 8-day composites using single day in situ data, and how do you compare those for validation? In a 8-day composite, which is not an 8-day average, you do not really know which day a given pixel stems from - or did you use this information?

- The Webster validation dataset which is your only independent validation dataset is either a typo or just wrong, there is no such dataset in that paper. Please double check. Line 149-152: You state that the validation data by Webster et al. 2015 supposedly stems from 2000-2014 and has resolution 8 to 25km2. When I look into that manuscript I discover that the study by Webster et al. 2015 is a fine approach to classify optical GFL images of 1m resolution, using collocated APLIS 2011 field campaign data - please see Table 1 in Webster et al 2015 for a list of the used data. These data are from 2011 only and have spatial resolution of 1 meter. I cannot find any other data in the manuscript by Webster et al 2015 which stems from 2000-2014 and has resolution 8-25km2.

- a scatter plot with "original MPF" and "retrieved MPF" on the axes, where each data point of the training and validation datasets can be seen, as well as the size of the sample, also for the Webster dataset. Your Fig. 4 cannot be used as the validation plot.
- make sure to use the original MODIS resolution and the finest spatial resolution of in situ data, both datasets also temporally collocated, to ensure a good quality of the comparison. For the transparency, it would be a good idea to provide case studies where you plot e.g. reference aerial values on your retrieved MODIS MPF map and discuss the discrepancies."

**Response:**

Based on the reviewer's comments, in this revision, we added more detailed descriptions about data and methods used in the manuscript.

1) We provided more information about the MODIS data used in this study, which is the MODIS/Terra Surface Reflectance 8-Day L3 Global 500m SIN Grid V006 (MOD09A1 version 6, https://lpdaac.usgs.gov/products/mod09a1v006/, Vermote, 2015). Note that the MODIS data used in Rösel et al. (2015) is the MOD09A1 version 5. Four spectral bands of MOD09A1 version 6 were used in our study (as the input data to the deep neural network, see section 4) for details) to derive MPF, including band 1, bandwidth of 620-670 nm; band 2, bandwidth of 841-876 nm; band 3, bandwidth of 459-479 nm; band 5, bandwidth of 1230-1250 nm. The improvements of MOD09A1 version 6 include: "a) Improvements to the aerosol retrieval and correction algorithm along with new aerosol retrieval look-up tables; b) Refinements to the internal snow, cloud, and cloud shadow detection algorithms. Uses Bidirectional Reflectance Distribution Function (BRDF) database to better constrain the different threshold used; c) Processes ocean bands to create a new Surface Reflectance Ocean product and provides Quality Assurance (QA) datasets for these bands; d) Improved discrimination of salt pans from cloud and snow, along with the inclusion of a salt pan flag in the QA band." (https://lpdaac.usgs.gov/products/mod09a1v006/). The MOD09A1 version 6 has a spatial resolution of 500 m and is available at 8-day interval. "Each pixel contains the best possible L2G (the Level 2G format, consisting of gridded Level 2 data, was developed as a means of separating geolocating from compositing and averaging) observation during an 8-day period as selected on the basis of high observation coverage, low view angle, absence of clouds or cloud shadow, and aerosol loading." (MODIS Surface Reflectance User's Guide,

https://lpdaac.usgs.gov/documents/306/MOD09\_User\_Guide\_V6.pdf ). According to the MODIS Surface Reflectance User's Guide Collection 6, each orbit observation is assigned a score, based on whether it is flagged for cloud, cloud shadow, high aerosol or low aerosol, or contains high view angle or low solar zenith angle. The lowest score, 0, is assigned to observations with fill values for data. The remaining scores are:

- 1 BAD: data derived from a faulty or poorly corrected L1B pixel
- 2 HIGHVIEW: data with a high view angle (60 degrees or more)
- 3 LOWSUN: data with a high solar zenith angle (85 degrees or more)
- 4 CLOUDY: data flagged as cloudy or adjacent to cloud
- 5 SHADOW: data flagged as containing cloud shadow
- 6 UNCORRECTED: data flagged as uncorrected
- 7 CLIMAEROSOL: data flagged as containing the default level of aerosols
- 8 HIGHAEROSOL: data flagged as containing the highest level of aerosols
- 9 SNOW: data flagged as snow
- 10 GOOD: data which meets none of the above criteria

The observation with the highest score and the lowest view angle is selected for the MOD09A1, which minimizes the effect of the clouds on the spectral reflectance.

**2)** We provided more information about the in-situ data used in this study. The observed MPF relative to grid (or image area) from six different sources (HOTRAX, DLUT, TransArc, PRIC-Lei, NSIDC, and NPI) were used in our study (as the target data in the deep neural network, see **section 4)** for details).

- HOTRAX: MPFs were collected during the Healy Oden Trans-Arctic Expedition (HOTRAX) by the Polar Science Center, University of Washington (Perovich et al., 2009). The HOTRAX was conducted from August to September 2005 to obtain physical properties of the ice pack. The cruise started from Alaska, crossed the Bering, Chukchi, and Beaufort Seas and the Arctic Ocean reaching the North Pole, and then headed south and exited the Arctic basin through Fram Strait. The ice survey was made based on ice station measurements, helicopter survey flights, and the deployment of autonomous ice mass balance buoys. Fractional areas of melt ponds were estimated during the expedition. The MPFs from HOTRAX used in the network training were measured at 77°-79°N and 84°-87°N on 13, 21, 29 August and 6 September 2005. The coverage of each MPF measurement is about 57 m×70 m. The obtained measurement from HOTRAX is the MPF relative to the grid (the coverage of each observation). The data can be found at http://psc.apl.uw.edu/data/.
- DLUT: MPFs were collected during two Chinese Arctic Research Expeditions by • the Dalian University of Technology (DLUT, Lu et al., 2010; Huang et al., 2016). The first survey of DLUT was conducted from July to September 2008 during the third Chinese Arctic Research Expedition. During the cruise, eight helicopter flights were conducted and more than 9000 aerial images were obtained in the Pacific sector of the Arctic. The MPF was estimated from the digital image with a camera resolution of 3264×2248 pixels. The flight altitude generally varied around 100 m according to weather conditions. At this height, each snapshot covers an area of approximately 98 m×67 m (Lu et al., 2010). The second survey of DLUT was conducted from July to September 2010. The underway ship- and helicopterbased ice observations were primarily in the Chukchi Sea, Beaufort Sea, Canada Basin and Central Arctic Ocean. The images were classified into three distinct surface categories (sea ice/snow, water and melt ponds). The areal fraction of each category is determined by a camera resolution of 3264×2248 pixels. The flight altitude varied between 150 m to 500 m. Each image covers an area between 147  $m \times 100$  m and 490 m  $\times 335$  m (Huang et al., 2016). The images from the two cruises are spaced without overlapping, and each image represents an independent scene. The DLUT MPF used in the network training was measured at 84°N and 86°N on 20 August 2008 and 13 August 2010. The coverage of each MPF estimated from the airborne image is about 98 m×67 m (first survey), 147 m×100 m and 490  $m \times 335$  m (second survey). The obtained measurement from DLUT is the MPF relative to the grid (image area).
- TransArc: MPFs were collected from the ice breaker RV Polarstern during the Germany Trans-Polar cruise ARKXXVI/3 (Nicolaus et al., 2012, hereafter referred to as TransArc). The TransArc conducted from August to October 2011. Visual observations of sea ice conditions were performed hourly from the bridge of

Polarster. Sea ice type and thickness, snow depth, pond coverage, and surface scattering layer depth were recorded during the cruise. The observations followed the ASPeCT protocol with additional observations on melt ponds. It should be noted that the TransArc MPF was recorded on multiyear and first-year ice respectively for some cases, the MPF was estimated by using the linear mix of these values. The recording visibility in TransArc ranges between 50 m to 10 km based on ASPeCT (https://epic.awi.de/id/eprint/31658/14/ASPeCt metcodes.pdf). The TransArc MPF used in the network training was measured at 84°-87°N on 13, 29 August and 6 September 2011 and the visibility of the records ranges from 500 m to 1000 m. The obtained measurement from TransArc is the MPF relative to sea ice cover The data can be found at https://doi.pangaea.de/10.1594/PANGAEA.803312.

- PRIC-Lei: MPFs were collected during the Arctic Research Expeditions by the Polar Research Institute of China in summer from 2010 to 2016 (Lei et al., 2017, hereafter referred to as PRIC-Lei). Half-hourly Arctic Shipborne Sea Ice Standardization Tool (ASSIST) observations were conducted at the bridge of the R/V Xuelong to document sea ice concentration, sea ice and snow thickness, fractions of melt ponds (the area ratio relative to sea ice cover), dirty ice (with severe impurity depositions) and ridging, and floe size. Sea ice concentration was only assessed for a local area with a diameter of 2 km, which was reduced to 1 km on foggy days and melt pond fraction was estimated around the ship within 1 km. The MPF of PRIC-Lei used in the network training was measured at 73°-88°N on 28 July, 5 and 21 August 2010, 79-84°N on 12, 20, 28 August 2012, 73-79°N on 5, 13, 29 August 2014 and 73-80°N on 27 July, 4 and 20 August 2016. The coverage of each MPF record is about 1 km×1 km. The measurement from PRIC-Lei is MPF relative to sea ice cover.
- NSIDC: The MPFs were obtained from the NSIDC during summer of 2000 and • 2001 (Fetterer et al., 2008). Development of this data set was based on experience gained using reconnaissance imagery during the Surface Heat Budget of the Arctic Ocean (SHEBA) and earlier summer ice monitoring experiments (NSIDC 2000, Fetterer and Untersteiner 1998). Visible band imagery from high-resolution satellites were acquired over the Beaufort Sea, the Canadian Arctic, the Fram Strait, and the East Siberian Sea during summer of 1999, 2000 and 2001. Imagery was analyzed using supervised maximum likelihood classification to derive either two (water and ice) or three (pond, open water, and ice) surface classes. The estimated pond coverage was under 500 m square cells within 10 km square images (image resolution is 1 m). The NSIDC MPF used in the network training was estimated from June to August in 2000 and 2001. The coverage of each MPF estimation is 500 m×500 m. The measurement from NSIDC is the MPF relative to grid (the coverage of each observation). The data can be found at https://nsidc.org/data/G02159/versions/1.
- NPI: The MPFs were collected by the Norwegian Polar Institute (NPI) during the field campaign on Arctic sea ice north of Svalbard in summer 2012 (Divine et al., 2015; Divine et al., 2016). The data set presents regional scale of about 150 km

morphological properties of a relatively thin, 70-90 cm modal thickness, first-year Arctic sea ice pack in an advanced stage of melt. The data comprises fractions of three surface types (bare ice, melt ponds, and open water) along the flight tracks calculated from images acquired by a helicopter-borne camera system during ice-survey flights from late July to early August 2012. For a typical flight altitude of about 35 m over sea ice, the camera lenses used in the setup provide a footprint of about 60 by 40 m. For typical helicopter roll (pitch) angles of about  $-2^{\circ}$  (1°), the distortion of the image plane from an ideal rectangular one and the associated uncertainty in the image area of less than 1% is considered insignificant. Therefore no correction for pitch and roll was applied to the images (Divine et al., 2015). The NPI MPF used in the network training was measured at 80-82°N on 3 August 2012. The coverage (footprint) of each MPF record is about 60 m×40 m. The obtained measurement from NPI is the MPF relative to sea ice. The data can be found at https://data.npolar.no/dataset/5de6b1e4-b62f-4bd4-889c-8eb7bb862d3b.

Figures 1 to 6 show the observed MPF (used as target data in the network training) in the original resolution from above six sources overlaid on the NASA Team sea ice concentration (SIC). The MPF here is the fraction relative to the grid area. It appears that most of the MPF observations are in the grid with SIC above 40%.

Figure 1. Observed MPF from HOTRAX overlaid on NASA Team SIC.